

# Past and future of the Arctic sea ice in HighResMIP climate models

Julia Selivanova[1,2], Doroteaciro Iovino[1], Francesco Cocetta[1]

[1] Ocean Modeling and Data Assimilation Division, Centro Euro-Mediterraneo sui Cambiamenti Climatici, Bologna, Italy

[2] University of Bologna, Department of Physics and Astronomy, Bologna, Italy

*Correspondence to*: Julia Selivanova (julia.selivanova@cmcc.it)

**Abstract.**

We examine the past and projected changes in Arctic sea ice properties in 6 climate models participating in the High Resolution Model Intercomparison Project (HighResMIP) in the Coupled Model Intercomparison Project Phase 6 (CMIP6). Within HighResMIP each of the experiments are run using a reference resolution configuration (consistent with typical CMIP6 runs) and higher resolution configurations. The role of horizontal grid resolution in both the atmosphere and ocean model components in reproducing past and future changes in the Arctic sea ice cover is analysed. Model outputs from the coupled historical (hist-1950) and future (highres-future) runs are used to describe the multi-model, multi-resolution representation of the Arctic sea ice and to evaluate the systematic differences (if any) that resolution enhancement causes. Our results indicate that there is not a strong relationship between the representation of sea ice cover and the ocean/atmosphere grid: the impact of horizontal resolution depends rather on the examined sea ice characteristic and the model used. However, the refinement of the ocean grid has a more prominent effect compared to the atmosphere: eddy-permitting ocean configurations provide more realistic representations of sea ice area and sea ice edge. All models project substantial sea ice shrinking: the Arctic loses nearly 95% of sea ice volume from 1950 to 2050. The model selection based on historical performance potentially improves the accuracy of the model projections and predicts the Arctic to turn ice-free as early as in 2047. Along with the overall sea ice loss, changes in the spatial structure of the total sea ice and its partition in ice classes are noticed: the marginal ice zone (MIZ) dominates the ice cover by 2050 suggesting a shift to a new sea ice regime much closer to the current Antarctic sea ice conditions. The MIZ-dominated Arctic might drive developments and modifications of model physics and parameterizations in the new generation of GCMs.

## 1 Introduction

Sea ice is the key feature of high-latitude climate through its role in the surface energy budget, ocean and atmosphere dynamics, and marine ecosystems. Over the recent decades, the Arctic has witnessed unprecedented sea ice loss, which is a



key indicator of global climate change (e.g. Onarheim et al., 2018; Serreze and Meier, 2019), driven both by anthropogenic
activities and internal climate variability (e.g. Notz and Stroeve, 2016). Arctic sea ice has declined in every month of the year
with the strongest trends in September, a sea ice extent (SIE) reduction of 79000 km$^2$ yr$^{-1}$ in the period 1979-2022, and in
March, with -39200 km$^2$ yr$^{-1}$ over 1979-2022 (http://nsidc.org/arcticseaicenews/2022/). The overall decrease in SIE reveals
large seasonal and regional variability. Although winter sea ice loss is dominated by the reduction in the Barents Sea (Årthun
et al., 2021), the most pronounced summer sea ice decrease occurs in the East Siberian Sea (that explains more than 20% of
the September trend, (Watts et al., 2021) and in the Beaufort, Chukchi, Laptev and Kara seas (Onarheim et al., 2018). Along
with a severe reduction in sea ice coverage, Arctic sea ice has also thinned, with a ∼70% reduction in summer sea ice volume
(SIV) over 1979-2021 (https://nsidc.org/). As a consequence, the Arctic ice is getting younger: the portion of the multi-year
ice, which previously was the iconic feature of the Arctic, has decreased from ∼30% in 1985 (beginning of the satellite era)
to ∼4.4% in 2020 in winter months (Perovich et al., 2020). The Arctic transition toward a first-year ice regime might
substantially alter the interactions in the ocean-atmosphere-ice system (Aksenov et al., 2017). The changes in total SIE and
sea ice thickness (SIT) cause redistribution of the sea ice classes, in particular the marginal ice zone (MIZ) is strongly affected
(Rolph et al., 2020). The Arctic MIZ has held interest as the fundamental region supporting many physical, biological and
biogeochemical processes (Tàpias et al., 2021). The MIZ is traditionally defined as the region where polar air, ice, and water
masses interact with the ocean temperature and subpolar climate system (Wadhams and Deacon, 1981). It corresponds to the
portion of the ice-covered ocean often characterised by highly variable ice conditions, where surface gravity waves
significantly impact the dynamics of sea ice (e.g. Dumont et al., 2011). Due to the large uncertainties in observed and forecasted
waves within sea ice, the MIZ is still operationally defined through a sea ice concentration (SIC) thresholds, as the transition
zone between open water and consolidated pack ice, where the total area of ocean is covered by 15-80% of sea ice (e.g. Strong,
2017; Paul et al., 2021; Rolph et al., 2020). While there are no significant changes in the area of the Arctic MIZ during the
satellite era (Rolph et al., 2020), the marginal ice zone fraction (MIZF) defined as the percentage of total sea ice area (SIA)
covered by MIZ (Horvat, 2021) increases by more than 50% in August and September as the total SIA drastically decreases
(Rolph et al., 2020; Horvat, 2021). Since the MIZ differs from the pack ice in higher sensitivity to the dynamic and
thermodynamic forces, the growing MIZF changes the Arctic response to global warming, which may worsen the pace of sea
ice melt and pose repercussions for local and global climate.
Assuming that the Arctic Ocean will continue to lose sea ice, a relevant question is how fast the Arctic will turn ice-
free in summer. Coupled climate models can be used in the prediction and projection of the climate system, including the sea
ice conditions. In the majority of simulations from CMIP6 (Eyring et al., 2016), the Arctic Ocean becomes practically sea ice
free (SIA < 1 million km$^2$) in September for the first time before 2050 in all scenarios (Notz and SIMIP Community, 2020) or
even by 2035 when selecting only the models that best represent the present Arctic sea ice state and northward ocean heat
transport (Docquier and Koenigk, 2021). Even using a process-based selection criterion, uncertainties in the model projections
are relatively large, which undermines the model's trustworthiness (Docquier and Koenigk, 2021). Besides, the accurate
simulation of past and present Arctic sea ice is still challenging. Although the CMIP6 multi-model ensemble mean is closer to



the observed sensitivity of Arctic sea ice to global warming (Notz and SIMIP Community, 2020; Shu et al., 2020), there is little difference in overall model performance among CMIP3, CMIP5 and CMIP6. CMIP6 models still simulate a wide spread of mean sea ice area and volume in March and September (Davy and Outten, 2020; Notz and SIMIP Community, 2020; Watts et al., 2021).

Among the model developments and improvements needed to produce more accurate future projections, the increase in horizontal spatial resolution is recognized to be a key step to enhance the representation of the complex processes at high latitudes and to obtain trustworthy projections of ice variability. In order to address the impact of the model grid resolution on the simulated oceanic and atmospheric phenomena, the High Resolution Model Intercomparison Project (HighResMIP; Haarsma et al., 2016) was designed within the EU Horizon 2020 PRIMAVERA project (PRocess-based climate sIMulation: AdVances in high-resolution modelling and European climate Risk Assessment, https://www.primavera-h2020.eu/). HighResMIP is one of the CMIP6-endorsed model intercomparison projects, which provides a useful framework to investigate the role of the enhanced horizontal resolution in representing the features of the climate system. A number of climate modelling groups contributed to the project providing the same simulations in at least two different configurations. The impact of the increased resolution within the HighResMIP is examined in many studies with regard to atmosphere, sea ice, and ocean components of the climate systems (e.g., Fuentes-Franco and Koenigk, 2019; Docquier et al., 2019; Bador et al., 2020; Roberts et al., 2020; Jackson et al., 2020; Lohmann et al., 2021; Meccia et al., 2021). Despite the fact that high-resolution models can resolve specific dynamical features, the role of the enhanced horizontal resolution is not uniform across ocean regions and models. Grist et al. (2018) demonstrated that refining the ocean grid to eddy-permitting resolution raises the Atlantic meridional heat transport and improves the agreement with observational estimates - they also show the significantly smaller impact of atmosphere resolution on the strength of the heat transport. Docquier et al. (2019) confirmed this finding and showed that a better representation of Atlantic surface characteristics, velocity fields, and sea surface temperature (in addition to transports toward the Arctic) improves the representation of the Arctic SIA and SIV. Nevertheless, the role of ocean resolution in the representation of ocean heat transport (OHT) and SIA is less clear when considering the regional effect on specific Arctic sectors, as shown for the Barents Sea in Docquier et al. (2020).

Here, we focus on the impact of horizontal resolution on the Arctic sea ice properties in the past and future at hemispheric and regional scales using the model outputs from coupled historical (hist-1950) and future (highres-future) runs from HighResMIP. We assess seasonal and interannual variability and trends in the SIA and SIV, and examine when the Arctic will see its first ice-free summer. We aim to explore the role of enhanced ocean/atmosphere horizontal resolution in the representation of past and current sea ice and to provide some insight into whether the grid refinement improves the model performance in predicting the future Arctic sea ice conditions.





**2 Data**

In this study, we analyse the outputs from the six coupled climate models participating in the HighResMIP. We use coupled runs with historical forcing (hist-1950) covering the period 1950-2014 and future projections (highres-future) from 2015 to 2050 based on the Fossil-fueled development SSP5 scenario. For the past sea ice properties, we mainly focus on the time period from 1979 to compare model results with available satellite records. For the ocean, five models use the Nucleus for European Modelling of the Ocean framework (NEMO, Madec et al., 2016), yet different versions, whereas MPI-ESM is based on the Max Planck Institute Ocean Model (MPIOM, Jungclaus et al., 2013). The basic characteristics of the models are given in Table 1. Because each of the models uses at least two different resolutions, we evaluate 14 configurations in total. CMCC-CM2 and MPI-ESM use one ocean (eddy-permitting) resolution with two different atmospheric grids. ECMWF-IFS and EC-Earth3P run two of three configurations with an eddy-permitting ocean and different atmosphere resolutions. In other models, ocean and atmosphere resolutions vary in concert among configurations. Note that ECMWF-IFS and HadGEM3 provide several ensemble members, however we use only the first ensemble member in this study; ECMWF-IFS is not considered in the analysis of future projections since it does not provide the outputs from highres-future experiments.

**Table 1. Models and specifications of their configurations used in the study.**

| Model configuration | | nominal ocean resolution (°) | nominal atmosphere resolution (km) | model components | |
|---|---|---|---|---|---|
| | | | | ocean-sea ice | atmosphere |
| CMCC-CM2 (Cherchi et al., 2019) | HR | 0.25 | 100 | NEMO3.6+CICE4.0 | CAM4 |
| | VHR | 0.25 | 25 | | |
| CNRM-CM6-1 (Voldoire et al., 2019) | LR | 1 | 250 | NEMO3.6+GELATO6 | ARPEGE6.3 |
| | HR | 0.25 | 100 | | |
| ECMWF-IFS (Roberts et al., 2018) | LR | 1 | 50 | NEMO3.4+LIM2 | IFS cycle43r1 |
| | MR | 0.25 | 50 | | |
| | HR | 0.25 | 25 | | |



| | | | | | |
|---|---|---|---|---|---|
| EC-Earth3P (Haarsma et al., 2020) | LR | **1** | **100** | **NEMO3.6+LIM3** | **IFS cycle36r1** |
| | HR | **0.25** | **50** | | |
| HadGEM3 (Williams et al., 2018) | LM | **1** | **250** | **NEMO3.6+CICE5.1** | **UM** |
| | MM | **0.25** | **100** | | |
| | HM | **0.25** | **50** | | |
| MPI-ESM (Müller et al., 2018) | HR | **0.4** | **100** | **MPIOM1.6.3** | **ECHAM6.3** |
| | XR | **0.4** | **50** | | |


The simulated SIA is validated against satellite observations. We use monthly SIC from two satellite-based products:
the NOAA/NSIDC Climate Data Record (version 4, Meier and Stewart., 2021, hereafter CDR) and EUMETSAT OSISAF
Climate Data Record and Interim Climate Data Record (release 2, products OSI-450 and OSI-430-b, Lavergne et al., 2019)
both for the period 1979-2021. CDR uses gridded brightness temperatures in low frequencies from the Nimbus-7 SMMR (18,
37 GHz) and the DMSP series of SSM/I and SSMIS passive microwave radiometers (19.4, 22.2, 37 GHz). Different ratios of
frequencies are used to filter weather effects. The output data are distributed on a 25 km x 25 km polar stereographic grid.
CDR algorithm blends the NASA Team (NT; Cavalieri et al., 1984) and the Bootstrap (BT; Comiso, 1986) by selecting the
higher concentration value for each grid cell, so taking advantage of the strengths of each algorithm to produce concentration
fields that are more accurate than those from either algorithm alone (Meier, 2014). OSISAF comprises two SIC products based
on passive microwave sensors: OSI-450 (from 1979 to 2015) and OSI-430-b, extension from 2016 onwards. OSI-450 uses
data from the SMMR 1979-1987), SSM/I (1987-2008), SSMIS (2006-2015) instruments (19.35 and 37 GHz frequencies)
together with Era Interim reanalysis (Dee et al., 2011), while OSI-430-b is based on SSMIS and operational analysis and
forecast from ECMWF. We use estimates of SIT and SIV from the Pan-Arctic Ice Ocean Modeling and Assimilation System
(PIOMAS; Zhang and Rothrock, 2003) that comprises the global Parallel Ocean and sea Ice Model (POIM) coupled to eight-
category thickness and enthalpy distribution sea ice model and a data assimilation of SST (from NCEP/NCAR reanalysis,
Kalnay et al., 1996) and SIC (from the NSIDC near-real time product; Brodzik and Stewart, 2016). PIOMAS proved its
credibility against in-situ measurements (Stroeve et al., 2014; Wang et al., 2016) and therefore it is widely used in numerous
intercomparison studies as the observational proxy (e.g. Labe et al., 2018). Note that PIOMAS tends to underestimate the thick





ice North to Greenland and the Canadian Arctic Archipelago and underestimate SIT in the areas of thin ice (Stroeve et al.,
2014; Wang et al., 2016). Monthly fields of SIC and effective SIT from 1979 to 2021 are used in this work. We describe sea
ice coverage in terms of SIA (the integral sum of the product of ocean grid-cell areas and the corresponding sea ice
concentration), instead of SIE (the integral sum of the areas of all grid cells with at least 15% of SIC). To compute SIV, the
equivalent SIT (the sea ice volume per grid-cell area) is multiplied by the individual grid-cell area, and then summed over the
Arctic region. To derive integrative metrics, only the grid cells with at least 15% SIC are considered owing to the high
uncertainty in passive microwave retrievals in low sea ice conditions. Apart from model evaluation at the hemispheric scale,
we provide a regional analysis of sea ice variability in six subregions of the Arctic Ocean as defined in Figure 1.
**3 Results**
**3.1 Mean state**
First, we assess the spatial patterns of simulated ice properties against observational-based estimates over the
historical period restricted from 1979 to 2014. Figure 2 shows the climatological mean distribution of SIT in March and
September for model outputs and PIOMAS. The mean position of 15% and 80% SIC edges is also shown from each model
and CDR (over PIOMAS). In general, most models struggle to reasonably simulate the spatial pattern of SIT and produce
either thicker (ECMWF-IFS, EC-Earth3P, CMCC-CM2 VHR4) or thinner (CNRM-CM6, MPI-ESM) ice over a vast area
compared to PIOMAS. Some models are able to correctly locate the thickest ice north of Greenland and the Canadian Arctic
Archipelago and the thinner ice in the Siberian Shelf Seas (HadGEM3, CMCC-CM2 HR4), but the simulated ice can thicken
up to 7 m. EC-Earth3P HR and ECMWF-IFS MR, despite capturing the overall SIT pattern, simulate high thickness also in
the East Siberian and Chukchi Seas, which is clearly visible in March. This might be related to unrealistic sea ice drift. As in
PIOMAS, most models reproduce changes in the SIT between March and September with a more pronounced seasonal retreat
in the Siberian sector.
There is no direct effect of horizontal resolution on the spatial distribution of SIT. Increasing ocean resolution, the
mean SIT decreases for ECMWF-IFS, does not change significantly for HadGEM3 and CNRM-CM6, and increases for EC-
Earth3P. The role of atmosphere resolution also depends on the model: for example, the finer atmosphere resolution MPI-ESM
reproduces on average slightly thinner ice compared to LR configuration, while the finer CMCC-CM2 simulates thicker ice
over a larger area. Biases in the representation of SIT pattern can be related to poor representation in surface pressure and
large-scale atmospheric patterns (Kwok and Untersteiner, 2011; Stroeve et al., 2014), sea ice motion and ocean forcing (Watts
et al., 2021).
Most models tend to realistically simulate the position of the sea ice edge both in March and September.
Configurations with finer ocean resolution have a better fit to CDR in the location of the 15% SIC ice edges. The LR
configuration of ECMWF-IFS tend to overestimate the sea ice cover far south in the North Atlantic and the North Pacific
Oceans compared to CDR. The bias can be explained by the poor representation of the ocean advection. In fact, Docquir et al.



(2019) showed that the northward OHT is improved when ocean resolution increases from 1° to 0.25°, both across the Bering
Strait (83 km wide) and through the Nordic Seas establishing the Atlantic warm inflow into the Arctic Ocean. Similarly, as for
SIT, the effect of the atmospheric grid resolution on the sea ice extent is model dependent. When it is enhanced, there are no
notable changes in the location of March ice edge in the ECMWF-IFS and HadGEM3 models, while it is largely overestimated
in CMCC-CM2 and MPI-ESM, particularly in the Nordic Seas. Specifically, CMCC-CM2 HR4 underestimates March sea ice
coverage in the northern Barents Sea, the Bering Sea, and the Sea of Okhotsk, whereas the VHR4 version (with finer
atmospheric grid) reproduces a reasonable amount of winter ice in marginal seas. In September, higher atmosphere resolution
leads to a larger SIA in ECMWF-IFS and CMCC-CM2, conversely it has an opposite effect in HadGEM3 and MPI-ESM
models. In Addition, MPI-ESM XR does significantly melt sea ice in the Siberian seas which are almost ice-free in summer.
The width of the MIZ (marked in Figure 2 by the area capped between 15% and 80% SIC contours) also varies among different
models. In many of them, March MIZ similarly surrounds the inner ice pack, comparing well with CDR. In September, most
models fairly simulate an extension of MIZ comparable to the observed one.  Exceptions are MPI-ESM runs that lose all
consolidated pack ice in summer and ECMWF LR that tends to overestimate the total and pack ice, with a small portion
covered by marginal ice in the Barents Sea and Nordic Seas.
**3.2 Seasonal variability**
Figure 3 shows the mean seasonal cycle of the total Arctic SIA and SIV computed over the 1979-2014 period. Satellite
estimates from both OSISAF and CDR are included to validate the models' outputs. The CDR Arctic ice area expands to its
maximum in March, with coverage of nearly $14 \times 10^6$ km$^2$, and returns to its minimum in September at around $6 \times 10^6$ km$^2$.
Similar seasonality is displayed by the OSISAF dataset, which has just a slightly smaller SIA in all months.
As in CMIP5 and CMIP6 low-resolution models (Shu et al., 2020, Notz and SIMIP Community, 2020), most HighResMIP
models adequately reproduce the mean seasonal cycle of SIA with the melt season starting in March and lasting until September
where a minimum is reached (Figure 3a). There is a considerable spread among models, it is relatively larger in winter than in
summer. March SIA ranges from 12 to $2010^6$ km$^2$, while September values lie in the range between 3 and $7.5 \times 10^6$ km$^2$ in all
but one model. The ECMWF-ISF LR overestimates the Arctic SIA all year round, but it can properly represent the amplitude
of SIA seasonal variability and hence correctly reproduces the ice advance and retreat phases. The comparison between the
model configurations indicates that finer resolution generally results in simulated SIA closer to satellite products. The effect
of changing atmosphere resolution varies among models, though. For instance, the CMCC-CM2 HR constantly stays in the
lower bound of the model ensemble and reproduces a weaker amplitude of the seasonal cycle compared to observations;
applying the atmospheric grid refinement (CMCC-CM2 VHR4 configuration) favourably increases sea ice coverage and does
not significantly change the seasonal cycle amplitude. Different impact is observed for the MPI-ESM model:  the finer
atmospheric grid leads to closer agreement with observations in SIA during winter but increases the spring/summer melting
resulting in underestimated September minimum up to ~50% compared to observations. In general, in other HighResMIP runs,
the atmosphere grid refinement gives smaller changes to Arctic sea ice coverage compared to the ocean resolution





enhancement. In the ECMWF-IFS, the LR shows a constant SIA overestimation, that is largely resolved in the model
configuration with an eddy-permitting ocean (HR), particularly in summer. As for the CMCC-CM2 model, a further refinement
in the atmosphere resolution increases the SIA in the whole year with the best agreement with observation from October to
July. The HadGEM3 runs are relatively close to observations in summer but they tend to overestimate the sea ice growth - the
impact of increased ocean and atmosphere resolution is evident for this model with a strong reduction of winter sea ice of
~25% from LL to HM and a smaller but still remarkable contraction in summer. Here, the increase in the atmosphere resolution
further reduces SIA in contrast to previous models. Finally, EC-Earth3P and CNRM-CM6 models show negligible differences
between model configurations, despite ocean and atmosphere grids resolution.

In our reference product, PIOMAS, the Arctic SIV ranges from $\sim 25 \times 10^3$ km$^3$ at its peak in April to $\sim 10 \times 10^3$ km$^3$ at

its minimum in August/September (Figure 3b). All models capture the timing of the SIV maximum in April and the minimum
in August/September with a realistic seasonal cycle amplitude that ranges between 15 and $20 \times 10^3$ km$^3$. However, there is a
large spread among different models , with most models overestimating PIOMAS - ECMWF-ISF LR is a clear "outlier"
exceeding $70 \times 10^3$ km$^3$ in April and $50 \times 10^3$ km$^3$ in September. Although in some models the bias in SIA is seasonally dependent
with larger errors in winter, bias in simulated SIV is consistent throughout the year in all models. In general, large SIV is
mainly due to poorly simulated SIT rather than uncorrect sea ice cover (Figure 2, 3a). Only in ECMWF-IFS LR, the
combination of large ice expansion and extremely thick ice leads to unrealistically high SIV. The SIV overestimation in the
CMCC-CM2 and EC-Earth3P models is caused by too thick sea ice, even though their SIA compare well with observations.
Only one model (CNRM-CM6 in both configurations) has thin ice and hence low bias in SIV compared to PIOMAS, all year
round. The changes in resolution have no visible impact in this case. The increase of only ocean resolution largely improves
the representation of SIV (as for SIA) in ECMWF-IFS with a large volume reduction, but does not affect the volume seasonality
in HadGEM3. Finer atmosphere resolution and the combined resolution increase tend to increase the ice volume except in
HadGEM3 and MPI-ESM. MPI-ESM has a good fit to PIOMAS for SIV although this model underestimates SIA and cannot
simulate consolidated pack ice (SIC > 80%, Figure 2).

In addition to the total SIA, we show the seasonal variability of the area covered by marginal ice over the same 1979-

2014 period (Figure 4a). First, it is worth noting that the evaluation of the simulated MIZ area is highly dependent on the
reference product used. It is worth noting the difference between CDR and OSISAF in the estimates of MIZ area, particularly
in summer. This can be mainly ascribed to the treatment of the wet surface (e.g. melt ponds, snow wetness) that poses difficulty
to retrieve the SIC using passive microwave radiometers (Ivanova et al., 2015). OSISAF has a small portion of MIZ in winter,
while it overestimates CDR from May to November. The maximum difference between the two products is up to nearly $0.9 \times 10^6$
km$^2$ in July. The observed MIZ seasonal variability contrasts with that shown by the total ice area: the MIZ expands in spring,
when the consolidated pack ice starts to melt, this process leads to the MIZ area peak occurring in summer. After reaching its
maximum in July, the marginal ice starts to melt and its area decreases until September, simultaneously with the total and the
consolidated pack ice cover. Before the next year's melting season, the MIZ stays relatively stable but with a secondary peak
in October, at the beginning of sea ice advance. The models are overall able to simulate the seasonal cycle, reasonably capturing





the phases of the MIZ expansion and retreat. However, they tend to overestimate the MIZ in winter, but most of them are lying
between the OSISAF and CDR summer estimates. Generally, models struggle to properly simulate the timing and magnitude
of the MIZ maximum: ECMWF-IFS LR is higher than observations from November to May due to a large overestimation of
the total ice area, nevertheless it lies between CDR and OSISAF in the rest of the year. Noteworthy, the ECMWF-IFS finer
resolution configurations are in better agreement with observed values. In the HadGEM3 LL configuration, the marginal ice
expansion starts earlier, with a large bias of the MIZ area from March to June. Increasing resolution in HadGEM3 model does
not have a visible impact for the rest of the year. The impact of changes in the ocean and atmosphere resolution is small for
other models. Finally, MPI-ESM configurations fail to reproduce the MIZ seasonal cycle from June to November. This pairs
with Figure 2, which shows underestimation of consolidated pack ice and MIZ predominance in the MPI-ESM runs.
We also show the seasonal cycle of the MIZ area fraction (MIZF) from 1979 to 2014, calculated from the model and
satellite products outputs (Figure 4b). The MIZF is defined as the percentage of the ice cover that is MIZ (Horvat, 2021) and
reflects the relative changes of the MIZ, which are highlighted since the total ice experiences substantial seasonal variability.
The observed MIZF ranges from 5-10% in winter to 20-40% at its maximum between June/July. For all models, the simulated
MIZF maxima are delayed compared to the satellite estimates and to the MIZ area by about one month, when the total ice area
approaches the September minimum and the MIZ area is still large. It is notable that the HighResMIP models are in better
agreement with observations when considering the MIZF rather than the MIZ area. Excluding the MPI-ESM configurations,
all models are in general agreement from November to May; the model spread enlarges in spring/summer but the models lie
anyway within the observation envelope. The use of the MIZF metric highlights the peculiar representation of Arctic sea ice
in the MPI-ESM: up to 95% of sea ice in the model consists of marginal ice.
**3.3 Seasonal variability in the sub-regions**
Since sea ice changes in the Arctic region are not uniform in space and time as a result of local climate effects (cf.
Parkinson et al 1999; Meier et al 2007, Peng and Meier 2018), it is important to monitor the sea ice change also on regional
scales. We analyse the seasonal variability of SIA and SIV in six sub-regions and we compare it with that of reference products
(Figure 5).
Satellite estimates of SIA are not shown in the Central Arctic sector (CA) due to the observation gap near the North
Pole. In this region, all models simulate a pronounced seasonal cycle in SIA with the widest area between December and April,
and a minimum in August. Although the majority of the models agree in winter when the region is fully covered by sea ice,
the inter-model spread increases in summer. HadGEM3 and CMCC-CM2 simulate similar seasonal cycles in all configurations
with slightly lower values in HadGEM3 HM. The ECMWF-IFS LR is an outlier also in this region, with a large SIA all year
round and a minimum in August that is as large as the autumn/winter values in other models. Also EC-Earth3P LR has SIA
comparable to ECMWF-IFS LR from November to May, however it overestimates the melting and growing phases with an
August minimum comparable to other models. The CNRM-CM6 model produces the smallest seasonal cycle amplitude in
both resolutions, with a decrease between the winter values and the minimum of ~10%. On the contrary, both MPI-ESM





configurations display the strongest seasonal cycle, with the largest area in winter and the smallest in summer. These
differences among models do not clearly depend on the resolution changes. For SIV, PIOMAS shows an increase of ~30%
between the minimum in August/September and the maximum in May. The seasonal cycle magnitude is captured by most
models but with a large spread mainly driven by differences in the simulated thickness (Figure 2). The models generally
perform similarly in simulating the SIV seasonal cycle in the sub-regions as at the hemispheric scale (Figure 3b). For the sake
of conciseness only the specific features of the SIV representation at the regional scale will be indicated below. The Barents-
Kara Seas (B-K) is the only sub-region where satellite products show a distinct maximum peak that occurs in April (one month
later the hemispheric SIA maximum), cf. Figure 5a. Except for CMCC-CM2, the models generally overestimate SIA in winter
with a large spread among them which reduces in summer, when models are in closer agreement with satellite estimates. The
strong underestimation of SIA in the CMCC-CM2 HR4 configuration could be attributed to the increased poleward Atlantic
OHT simulated by this model (Docquier et al., 2020). The warmer ocean temperatures not only promote sea ice melting in
winter but also hinder its growth in autumn. The ocean and atmosphere spatial resolution have generally the opposite effects
on simulated SIA. Increasing only the ocean resolution in ECMWF-IFS (from LR to MR) and HadGEM3 (from LL to MM)
results in lower SIA and a better fit to the observations. Conversely, increasing the atmosphere resolution generally leads to
larger SIA, except for decrease in SIA for HadGEM3. The combined effect of enhanced resolution in both ocean and
atmosphere in CNRM-CM6 and EC-Earth3P models increases the winter SIA, worsening the comparison with the
observations. For SIV, nearly a half of the model ensemble is within the 15% of PIOMAS seasonal variability from January
to June which is not the case for other sectors. The Barents-Kara Seas is the only region where CMCC-CM2 HR underestimates
SIV as a result of too low SIA. In addition, both configurations of CMCC-CM2 underestimate the seasonal variation of SIV.
At the same time, CNRM-CM6 has a better fit to PIOMAS SIV in the Barents-Kara Sea sector compared to the other parts of
the Arctic Ocean. The increased ocean resolution has a clear positive effect on SIV representation in ECMWF-IFS
configurations, whereas other models display similar values when changing such parameter. On the other hand, the enhanced
atmosphere resolution leads to higher SIV for ECMWF-IFS and CMCC-CM2, lower SIV for HadGEM3 and does not affect
SIV in MPI-ESM.
The Laptev (LV), East Siberian (ESS), and Beaufort-Chukchi Seas (B-C) show similar behaviour in SIA and SIV.
They can be analysed together and grouped as in Peng and Meier (2018). In these regions, there is no noticeable peak in the
observed seasonal variability of SIA, instead the annual maximum is extended between December and May since the winter
sea ice expansion is constrained by land. In spring, the downward shortwave radiation increases, causing the rapid sea ice melt,
which ends in September. Notably, the disagreement between satellite estimates in summer SIA is higher in all three regions
probably due to the enhanced presence of melt ponds, which complicate the SIC retrievals from passive microwave radiometers
(Ivanova et al., 2015). The models exhibit better agreement in winter, while the spread across models is larger in summer. This
can be associated with the model differences in simulating the river discharge (Park et al., 2020) as well as the transport of
Pacific waters through the Bering Strait (Watts et al., 2021), which modify the thermo-haline structure of the upper-ocean and
affect sea ice growth and melt. In all three regions, SIA from ECMWF-IFS LR is well compared with satellite estimates in





winter, which is not the case for other sectors with a greaterrole of the Atlantic OHT where the model is biased high. HadGEM3
overestimates SIA, particularly in its lower resolution configuration. This behaviour is common also for other parts of the
Arctic Ocean which points out that bias in HadGEM3 is similarly distributed across the regions. MPI-ESM underestimates
SIA with a greater degree in summer since the model is struggling to simulate consolidated pack ice (Figure 2). CNRM-CM6,
CMCC-CM2 and HR of EC-Earth3P show a fairly good agreement with satellite estimates in all three regions. Lower
resolution configuration of EC-Earth3P displays an earlier and faster sea ice retreat in the Laptev and East Siberian Seas
resulting in the second-lowest SIA, while the model compares well with OSISAF estimates in the Beaufort-Chukchi Seas.
Increased ocean resolution leads to lower SIA for all models except for EC-Earth3P which has higher values in its HR
configuration. The effect of the ocean resolution is stronger in summer, however the impact is substantial all year round for
HadGEM3. Enhancement of the atmosphere resolution does not significantly affect ECMWF-IFS but leads to higher summer
SIA in CMCC-CM2, as in the other regions. For MPI-ESM, the increase in atmosphere resolution has a larger impact on
summer SIA in the Laptev, East Siberian, and Beaufort-Chukchi Seas compared to other sectors: MPI-ESM XR simulates SIA
almost twice lower than CDR in August and September. In the Laptev, East Siberian, and Beaufort-Chukchi Seas, SIV reaches
the maximum in May (April-May in B-C) while the annual minimum occurs in September. Most models overestimate SIV
with the highest bias (ECMWF LR) in the East Siberian and Beaufort-Chukchi Seas. CMCC-CM2 HR and MPI-ESM HR are
the closest to PIOMAS, even though the latter fails to reasonably simulate the SIC (Figure 2). The effect of the ocean resolution
on SIV is clearly seen for ECMWF-IFS and EC-Earth3P in all three regions and for HadGEM3 in the Laptev Sea - the only
region where LL and MM configurations of HadGEM3 differ. Other models do not show considerable differences in SIV
when changing ocean resolution. Finally, increased atmosphere resolution results in higher SIV for ECMWF-IFS, EC-Earth3P,
and CMCC-CM2 and lower SIV for HadGEM3 and MPI-ESM.
The Greenland region (GD) holds the largest area of sea ice both in winter and summer (3 and $1.5\times10^6$ km²
respectively according to the satellite estimates). Most models tend to overestimate SIA all year round with the highest bias in
winter in ECMWF-IFS LR and HadGEM3. The models are generally capable of melting away the excess of sea ice by August,
so there is more consistency among most models in summer, when MPI-ESM underestimates SIA more than all of them. An
increase in the ocean resolution from 1° to 0.25° effectively improves the representation of SIA in ECMWF-IFS, whereas it
does not give notable changes in HadGEM3 and EC-Earth3P. The effect of atmosphere resolution again depends on the model.
ECMWF-IFS and CMCC-CM2 display slightly higher SIA in their finer atmosphere configurations, particularly in winter.
Conversely, HadGEM3 has lower SIA in its HM configuration in winter, which fits better to the observations. For MPI-ESM,
there are no differences between different configurations, as in the Barents-Kara Seas region. For SIV, both configurations of
CMCC-CM2 have a large error in the Greenland region owing to high bias in SIT (Figure 2); whilst at least one configuration
of the model is in good agreement with PIOMAS in other sectors. Enhanced ocean resolution leads to lower SIV for ECMWF-
IFS and higher SIV for EC-Earth3P. At the same time, there are no significant differences between configurations of
HadGEM3 and CNRM-CM6 with changing ocean resolution. An increase in the atmosphere resolution has almost no effect
on SIV in HadGEM3 and MPI-ESM but leads to higher SIV in CMCC-CM2



The displayed analysis reveals that the model performance and the accuracy of simulated SIA largely depend on the
Arctic region and the season studied. While Barents-Kara Seas and Greenland regions contribute mainly to the winter inter-
model spread, the largest summer differences among models are seen in the Laptev, East Siberian and Beaufort-Chukchi Seas.
There are no considerable differences in the model ability to simulate SIV at the regional scale, in fact the biases are generally
uniform across regions and seasons. Generally, we find no strong dependence of sea ice realism from the horizontal resolution.
The impact of the ocean resolution on the representation of SIA is most pronounced in the Barents-Kara Seas and Greenland
sea ice regions that are strongly influenced by the Atlantic OHT. The effect of the atmosphere resolution is less clear but there
is evidence that the atmosphere resolution has a stronger impact on SIV rather than on SIA and particularly in the regions of
thicker ice (B-C, GD).

**3.4 Interannual variability and trends**
Next, we evaluate the long-term variability of the Arctic SIA and SIV from the hist-1950 simulations from 1979 to
2014. Figure 6a illustrates monthly anomalies of SIA (with respect to 1979-2014 climatologies) simulated by the models and
derived from satellite data sets. The inter-model spread is relatively similar throughout the period but it increases from the
mid-2000s when the ice reduction has accelerated. All models are able to reproduce the sea ice shrinking but with varying
intensity: ECMWF-IFS LR, HadGEM3 LL, MPI-ESM HR show larger negative trends compared to observations (-44x10$^3$
km$^2$ yr$^{-1}$ in CDR and -46x10$^3$ km$^2$ yr$^{-1}$ in OSISAF), while the MR and HR versions of ECMWF-IFS, both configurations of
CNRM-CM6, EC-Earth3P, HadGEM3 HM, and CMCC-CM2 HR display weaker negative trends (Table 1). Nevertheless,
none of the models can capture the record lows of 2007 and 2012. An increase in the ocean resolution generally results in
smaller negative trends except for EC-Earth3P which shows a similar decline rate in both configurations. The effect of finer
atmosphere resolution is different among models: the SIA decrease is stronger in ECMWF-IFS and CMCC-CM2 and weaker
in HadGEM3 and MPI-ESM.
Figure 6b shows monthly anomalies of SIV (with the seasonal cycle removed) over 1979-2014 in HighResMIP
models and PIOMAS. There is a substantial inter-model spread for SIV compared to SIA, particularly at the beginning and the
end of the observed period (55-85% of yearly averaged SIV from PIOMAS). The biases from few models are not consistent
throughout the years varying significantly from positive to negative (EC Earth-3P HR, ECMWF MR, HadGEM3 LL).
Generally, models are in better agreement with reference product for SIV interannual variability compared to SIA (the
correlation coefficient for most models is higher than 0.75 for SIV against less than 0.2 for SIA). The weakest agreement is
found for ECMWF-IFS MR (R=0.28) and CNRM-CM6 (R=0.51 in LR and R=0.61 in HR). Increasing atmosphere resolution
results in a weaker correlation with PIOMAS (for HadGEM3, the correlation ranges from 0.91 (MM) to 0.82 (HM); for CMCC-
CM2, 0.93 (HR) and 0.87 (VHR); for MPI-ESM, 0.9 (HR) and 0.54 (XR)).
PIOMAS simulates sea ice shrinking at the rate of -291 km$^3$ yr$^{-1}$; similarly, all models simulate a SIV decrease. There
is no straightforward impact of changing resolution in ocean and atmosphere on the linear trends in SIV since the impact of
horizontal resolution on SIA and SIT differs with the models. However, we find that configurations with coarse ocean



resolution generally tend to simulate more negative trends (-424 km$^3$ yr$^{-1}$ in ECMWF LR compared to -105 and -157 km$^3$ yr$^{-1}$
in its finer configurations; for HadGEM3, the trend ranges from -355 km$^3$yr$^{-1}$ in lower resolution to -257 and -174 km$^3$ yr$^{-1}$ in
finer resolution configurations). Here, the exception is EC-Earth3P in which the eddy-permitting configuration has a larger
negative trend in SIV (-322 and -460 km$^3$ yr$^{-1}$). In CNRM-CM6, the SIV decrease is very weak (-62 and -36 km$^3$ yr$^{-1}$ for LR
and HR configurations, respectively), which might reflect the negative ice growth-ice thickness feedback: thin ice allows sea
ice to grow more rapidly mitigating the ice loss. The finer atmosphere resolution has different impact on the pace of sea ice
retreat in different models: CMCC-CM2, VHR4 and ECMWF-IFS HR simulate slightly stronger trend compared to their
coarser counterparts (-384 km$^3$ yr$^{-1}$ and -411 km$^3$ yr$^{-1}$ in CMCC-CM2; -105 and -158 km$^3$ yr$^{-1}$ in ECMWF-IFS). On the other
hand, in MPI-ESM and HadGEM3, the finer configuration has less negative trend compared to the coarser one (-337 km$^3$ yr$^{-1}$
and -144 km$^3$ yr$^{-1}$ in MPI-ESM; -174 and -257 km$^3$ yr$^{-1}$ in HadGEM3).
We also examine how the models simulate sea ice response to the external forcing on a seasonal scale. The monthly
trends in the Arctic-wide SIA (computed over the period 1979-2014) reveal that the models tend to underestimate the rate of
sea ice loss in the melting season and in summer (not shown). Most models reproduce more negative trends from November
to May and underestimate the magnitude of trends in other seasons. MPI-ESM HR trends are found to have a closer fit to the
observed trends for the total Arctic although the model is wrong in simulating SIC and sea ice classes. For SIV, the models
vary greatly in the representation of trends. Despite all models being able to simulate a SIV decline in all months, they cannot
capture the observed magnitude of sea ice loss and have values ranging from almost 0 to -450 km$^3$ yr$^{-1}$. They also struggle to
reproduce the seasonal cycle in the trend which in PIOMAS has a slightly stronger signal in June and a weaker signal in the
winter months (-320 km$^3$ yr$^{-1}$ and -260 km$^3$ yr$^{-1}$ respectively).

**Table 2. Linear trend in SIA and SIV and their standard deviations for 1979-2014 and 2015-2050 periods.**

| | 1979-2014 SIA trend (10$^3$ km$^2$/yr) | 2015-2050 SIA trend (10$^3$ km$^2$/yr) | 1979-2014 SIV trend (km$^3$/yr) | 2015-2050 SIV trend (km$^3$/yr) |
|---|---|---|---|---|
| ECMWF-IFR LR | -72.08 ± 16.9 | No future runs | -423.86 ± 68.3 | No future runs |
| ECMWF-IFR MR | -21.24 ± 9.8 | | -104.82 ± 71.4 | |
| ECMWF-IFR HR | -36.67 ± 7.6 | | -157.58 ± 34.4 | |
| EC-Earth3P | -34.2 ± 9.47 | -52.31 ± 16.1 | -322.28 ± 31.8 | -210.56 ± 64.1 |
| EC-Earth3P HR | -40.13 ± 8.8 | -54.87 ± 5.5 | -460.47 ± 97.5 | -368.47 ± 31.7 |
| CNRM | -29.83 ± 8.9 | -6.55 ± 13.4 | -61.89 ± 23.6 | -35.55 ± 26.7 |





| | | | |
|---|---|---|---|
| CNRM HR | -15.94 ± 7.9 | -63.9 ± 9.2 | -35.58 ± 15.9 | -131.21 ± 20.5 |
| HadGEM3 LR | -56.54 ± 13.1 | -113.91 ± 12.5 | -354.64 ± 66.2 | -361.87 ± 31.7 |
| HadGEM3 MM | -48.32 ± 10.8 | -97.68 ± 11.3 | -256.75 ± 41.2 | -459.86 ± 36.7 |
| HadGEM3 HM | -31.54 ± 8.3 | -106.72 ± 10.2 | -173.72 ± 38.5 | -440.09 ± 52.6 |
| CMCC-CM2 HR | -38.57 ± 5.2 | -47.55 ± 9.7 | -384.2 ± 30.9 | -286.38 ± 31.2 |
| CMCC-CM2 VHR | -40.83 ± 6.6 | -73.97 ± 6.6 | -411.1 ± 51.1 | -698.79 ± 37.5 |
| MPI-ESM HR | -52.19 ± 5.1 | -49.94 ± 8.3 | -336.95 ± 22.8 | -116.95 ± 19.7 |
| MPI-ESM XR | -36.94 ± 9.5 | -46.95 ± 8.5 | -143.97 ±44.5 | -99.39 ± 16.4 |
| CDR | -44.14 ± 7.3 | | | |
| OSISAF | -46.42 ± 6.7 | | | |
| PIOMAS | | | -291.27 ± 36.8 | |

Since there is a substantial difference in the models' performance in reproducing the seasonal variability on a regional scale, we analyse monthly trends in SIA and SIV in each sea ice zone over 1979-2014 (Figure 7). The magnitude and timing of sea ice loss strongly depend on season and region. According to observations, the winter decrease in SIA is most dramatic in the Barents-Kara Seas (nearly $-17 \times 10^3$ km$^2$ yr$^{-1}$; 0.8% yr$^{-1}$) while the summer trends are dominated by the Eastern Siberian Sea and Beaufort, and Chukchi Seas (almost $-25 \times 10^3$ km$^2$ yr$^{-1}$; 2-3% yr$^{-1}$). The Barents-Kara Seas and the Greenland region show a pattern of SIA trends that differs from the total Arctic and the rest of the regions which have one pronounced negative peak in September and trends close to zero in winter. Instead, in the Atlantic sector, i.e. Barents-Kara seas and Greenland coast, sea ice loss is observed all year round with a slightly stronger decrease in July. In the Central Arctic, the models simulate a weak SIA reduction with the strongest signal in August-September, which is not significant in most models (less than 5% of the SIA of the sector). In the other sectors, the models generally tend to underestimate the pace of sea ice loss indicated by satellite estimates. The exception is the Barents-Kara Seas and Greenland where some models produce more negative trends compared to the observations. In the Laptev, East Siberian, and Beaufort and Chukchi Seas some of the models do not simulate a reduction in summer SIA and even display weak positive trends, yet insignificant. Given that all these regions hold a large MIZF in summer (Figure 4), the inability to capture trends points to inaccurate sensitivity of sea ice to the external forcing, particularly within the MIZ.





The strongest negative trends in SIV are observed in the areas of thick ice: the Beaufort and Chukchi Seas (up to -90
$km^3 yr^{-1}$ in September), the Greenland sector (-80 $km^3 yr^{-1}$ in July), and the East Siberian Sea (-70 $km^3 yr^{-1}$ in summer months).
The seasonal cycle of the Barents-Kara Sea SIV trend contrasts with those of other sectors where the highest rate of sea ice
decline is observed in September. Notably, in the Laptev, East Siberian, and Beaufort and Chukchi Seas, SIV experiences a
substantial decrease in the winter months while SIA stays nearly stable reflecting a considerable ice thinning primarily driven
by basal melting. In the East Siberian Sea and Beaufort-Chukchi Seas, almost all models tend to underestimate trends in SIV
(10 out of 14 models produce less negative trends) while in the rest of the Arctic zones, PIOMAS is nearly in the middle of
inter-model spread. Compared to other models, both CNRM-CM6 configurations and the two finest configurations of
ECMWF-IFS have the changes in SIA and SIV closer to zero in almost all regions and months. On the one hand, CNRM-CM6
simulates very thin ice so the lack of trend is consistent with the concept of negative ice thickness-ice growth feedback. On
the other hand, ECMWF-IFS MR and HR underestimate sea ice reduction everywhere despite simulating very thick ice.
HadGEM3 performs differently at regional scale but at least one of the configurations has a very good fit to the PIOMAS
estimates. Generally, both configurations of CMCC-CM2 present the large SIV decrease in all sectors except for the Barents-
Kara Sea and the rate of decline is similar between two resolutions despite significant difference in the mean SIV. The HR
configuration of MPI-ESM is in a fairly good agreement with PIOMAS in all regions except the Central Arctic and the Laptev
Sea where it tends to produce more negative trends. Conversely, MPI-ESM XR underestimates negative SIV trends in all parts
of the Arctic Ocean except the Greenland zone where it is close to its HR configuration.
Overall, there is no consistent link between the strength of sea ice retreat and the ocean/atmosphere resolution, it
rather depends on the region and the model used. Considering only SIA, the models generally underestimate the trends
especially in finer ocean configurations and in Laptev, East Siberian and Beaufort and Chukchi Seas in summer. However,
beneficial effects of increased ocean resolution for SIA trends are observed for ECMWF-IFS in the Barents-Kara Seas and the
Greenland area. In these regions, other models do not considerably differ between configurations; low and high resolution
configurations show closer fit to the observations according to the season. Moreover, the increased atmosphere resolution also
does not improve the representation of SIA trends; HadGEM3, CMCC-CM2 and MPI-ESM finer atmosphere configurations
lead to underestimate the negative SIA trends more than their counterparts at coarse resolution. The relation between
ocean/atmosphere resolution and SIV trends is less clear and depends on the region and the model.


**3.5 Future projections**


In this section, we analyse the results of HighResMIP models when simulating future Arctic sea ice changes using highres-
future model outputs from 2015 up to 2050. HighResMIP future projections generally show a stronger sea ice loss compared
to historical runs (Table 2). These simulations can elucidate when the Arctic will reach its first "ice-free" summer, i.e. the
condition typically defined as the timing when September sea ice drops below $10^6$ $km^2$. Reaching ice-free conditions is an
unprecedented change in the Arctic environment and the tipping-point in the Earth's climate system. Considering the large



inter-model spread in simulating observed mean sea ice state and trends, we assume that a selection of the models which better
agree with observations can reduce the spread and decrease uncertainty in the model projections. We select models based on
their historical performance of September SIA and SIV mean state and trends against CDR and PIOMAS, respectively (Figure
8). To exclude outliers, we define the 75th percentile threshold and we select the models whose values do not exceed the
threshold for both variables. The resulting subset includes four models: low-resolution configuration of EC-Earth3P,
HadGEM3 MM and HM, and CMCC-CM2 HR. These models are used in the further analysis on sea ice future evolution.

Figure 9 illustrates the September SIV time series from 1950 to 2050 computed for total Arctic and sub-regions. The

vertical lines mark first ice-free September in the multi-model mean with and without model selection (yellow and green,
respectively) and in CDR (black, data available between 1971-2021). At the regional scale, the timing of ice-free conditions
refers to the threshold of 25% of the CDR SIA averaged over the 1980-2010 period in the given region. It is evident that huge
sea ice reduction takes place in all Arctic sectors, however the pace of sea ice loss varies across the regions owing to differences
in the initial state and dominant processes driving the change. We can note that applying model selection results in earlier
timing of the ice-free conditions in Barents-Kara, Laptev, East Siberian, and Beaufort-Chukchi Seas and in ice-free conditions
in the total Arctic, Central Arctic, and Greenland region. In latter sub-regions, multi-model mean without model selection does
not predict the event everywhere before 2050. The comparison between the model configurations in simulating timing of ice-
free conditions shows that there is no clear link between the model resolution and the pace of sea ice loss (not shown).
The September Arctic-wide sea ice from the multi-model mean (with model selection) shrinks by 95% from 1950 to 2050, cf.
top panel of Figure 9. The inter-model spread decreases throughout the century from $14\times10^3$ in 1950 to $1.64\times10^3$ km$^3$ in 2050.
The Arctic does not reach the ice-free conditions within 2050 in the multi-model mean without model selection, although
applying selection criteria advances the timing of the event up to 2047. The Central Arctic September sea ice loses 96% of its
volume by 2050 in the multi-model ensemble, which is in good agreement with PIOMAS in the overlapping period. The inter-
model spread again narrows substantially from $2.58\times10^3$ km$^3$ in 1950 to $0.23\times10^3$ km$^3$ in 2050. The ice-free conditions in the
Central Arctic are not reached before 2050 in the multi-model mean when considering all models. However, outliers' exclusion
leads to approaching the threshold in 2042. The Barents-Kara Seas experience the most dramatic sea ice loss accounting for
almost 100% of SIV from 1950 to 2050 in the models' ensemble. First ice-free September in the Barents-Kara Seas is
accurately simulated by the multi-model mean with model selection: the event occurs in 2012 as for CDR. Avoiding model
selection postpones the event by 19 years. In the Barents-Kara Seas, the spread among models is decreasing from $1.46\times10^3$
km$^3$ in 1950 to almost vanishing in 2050. The multi-model mean SIV in the Laptev Sea shrinks by 99% during 100 years. The
inter-model spread narrows from nearly $0.9\times10^3$ km$^3$ at the beginning of the run to $0.05\times10^3$ km$^3$ in the end. The timing of the
first ice-free summer is similar to that in the Barents-Kara Seas: SIA drops below the threshold in 2012 for CDR and in 2032
for the multi-model mean without model selection. When applying selection criteria, the ice-free conditions are reached in
2023. In the East Siberian Sea, September ensemble-mean SIV is reduced by 99% by the middle of this century. The East
Siberian Sea reaches the threshold in SIA earlier compared to the other regions. CDR produces the event in 2007, when the
Arctic broke the first record low while the multi-model mean with model selection simulates first ice-free conditions in 2033



(2034 without model selection). The inter-model spread ranges between $4.76 \times 10^3$ km$^3$ in 1950 and $0.1 \times 10^3$ km$^3$ in 2050. The
Beaufort-Chukchi Seas lose nearly 96% of SIV in 100 years in the ensemble-mean. The inter-model spread decreases from
$3.44 \times 10^3$ km$^3$ at the beginning to $0.37 \times 10^3$ km$^3$ at the end of the run. The multi-model mean reaches the first ice-free September
in 2046. When adopting the model selection, the Beaufort-Chukchi Seas are ice-free in 2039. The Greenland region is
undergoing the least prominent sea ice loss accounting for 88% throughout the period from 1950 to 2050. However, there is a
great narrowing of the inter-model spread from $6.12 \times 10^3$ km$^3$ in the middle of the last century to $1.15 \times 10^3$ km$^3$ 100 years after.
Both multi-model means project that Greenland SIA might turn ice-free in 2048. Overall, the models simulate the first ice-free
September later than CDR in all sub-region studied. Therefore, we can fairly assume the same behavour for the Total Arctic

Along with overall sea ice loss, there are substantial changes in the structure of sea ice cover. Figure 10 shows the

time series of September SIA and the MIZF from 1950 to 2050. For SIA (top panel), the models are in fairly good agreement
with the observations, yet have systematic biases and underestimate the negative trend. In addition, the inter-model spread is
large but relatively similar throughout the years ($\sim 4 \times 10^6$ km$^2$). For the MIZF (bottom panel), the spread among models
increases considerably with time from $\sim 10\%$ in 1950 to $\sim 75\%$ in 2050. Most models simulate the MIZF growth, which reflects
the transition of the sea ice state to the marginal ice-dominated. The MIZ in the 2040s is projected to account for up to 80% of
the total ice area in September, although the interannual variability at the end of the run is large in most models. CNRM-CM6
and MPI-ESM models are two outliers: CNRM-CM6 has a nearly constant MIZ fraction during the whole period, while MPI-
ESM has MIZF close to 100% from the beginning of the run but it occasionally drops to 0 at the end of the run. Distinct
models' performances in simulating MIZF show that an accurate representation of the total SIA does not guarantee the same
for all sea ice classes, highlighting the importance of studying the Arctic MIZ.
**4 Discussion**

Although the latest generation of the models does a fairly reasonable job in simulating the mean state and long-term

variability of sea ice cover (Notz and Community, 2020), the models still suffer from biases, which decrease the model's
trustworthiness in projecting the future sea ice state in the Arctic. The enhancement in the model components' horizontal
resolution is used in the CMIP6 HighResMIP as one of the factors capable of improving the realism of the model simulations
and reducing biases in polar regions. In this study, we investigated the ability of HighResMIP in simulating Arctic sea ice
variability and the impact of the ocean and atmosphere horizontal resolution on the representation of sea ice properties in the
recent past and future climate. We do not find a strong link between ocean/atmosphere resolution and the representation of sea
ice properties, and the realism of model performance rather depends on the model used. Nevertheless, there is evidence that
an enhanced ocean resolution leads to improved representation of winter SIA in some models. This is associated with a more
accurate meridional heat transport (Docquier et al., 2019) which is a key process that can regulate the location of the ice edge
and SIA (Li et al., 2017; Muilwijk et al., 2019). The Atlantic Ocean is the main heat source entering the Arctic, accounting for
73 TW on average per year (Smedsrud et al., 2010), therefore an adequate simulation of the boundary currents is particularly
important in the Atlantic sector of the Arctic Ocean which is confirmed by the regional analysis in our study. Another process
that might be sensitive to horizontal ocean resolution is the Arctic river discharge, which contributes both to seasonal variations



of sea ice cover and long-term sea ice variability. The freshwater input stabilizes the upper ocean stratification and isolates the warm Atlantic layer from the bottom of sea ice cover (Carmack et al., 2015), resulting in higher ice growth in winter. On the other hand, the heat input from the rivers accelerates sea ice melt and increases the ocean temperature, which has possible implications for the next year's growing season (Park et al., 2020). The representation of river discharge in HighResMIP models needs additional investigation. Our results do not show the systematic impact of atmosphere resolution on the representation of the Arctic sea ice. This is confirmed by other studies reporting the minor role of atmosphere resolution compared to that of the ocean (Roberts et al., 2020; Koenigk et al., 2021; Meccia et al., 2021). However, increasing atmosphere resolution might permit a more realistic representation of precipitation, which can lead to increased snowfall (Strandberg and Lind, 2021) and consequently invoke cooling and sea ice expansion (Bintanja et al., 2018).

SIT is less responsive to changes in the ocean grid resolution compared to SIA and its representation largely depends on the sea ice model. Our results show that in some cases large biases in SIT reduce the beneficial effect of increased horizontal resolution to SIA. Poor representation of SIT is a great obstacle to the robustness of sea ice projections. The high uncertainty cannot be overcome without constraining the model simulations with a sufficient number of in-situ measurements of the Arctic SIT, which are still sparse and unreliable (Massonnet et al., 2018). Apart from the horizontal resolution, there are other important factors affecting the model performance; for example, inaccurate representations of mixed layer depth (Watts et al., 2021), surface air temperature (Papalexiou et al., 2020), surface pressure and geostrophic winds (Kwok and Untersteiner, 2011; Stroeve et al., 2014), and sea ice sensitivity to global warming (Zhang, 2010). These elements pair with the intrinsic complexity of sea ice models that include thermodynamics schemes and parametrizations (Keen et al., 2021), sea ice dynamics components (Hunke, 2010) and coupling between the ocean and atmosphere components (Hunke et al., 2020). Given few improvements with increased horizontal resolution, we argue that running the models at higher resolution might not be worth the major effort of costly computations. Our results suggest that the efforts of the modelling groups should be aimed rather at the improvement of the sea ice model physics and parameterizations.

In this study, we try to understand when the Arctic will see its first ice-free summer using HighResMIP outputs. Models show a wide temporal range for the occurrence of ice-free conditions in the Arctic. To reduce the inter-model spread in sea ice projections we apply a widely used approach based on the selection of models according to their historical performance (Wang and Overland, 2012; Sentfleben et al., 2020). Although close agreement with observations do not guarantee the realism of the models, we believe that excluding the models that struggle to reproduce present-day SIA and SIV mean state and trends might improve the accuracy of future sea ice projections. Different criteria to select "best-performing" models exist and almost always lead to earlier near-disappearance of sea ice compared to no selection (Docquier and Koenigk, 2021). The timing of the first ice-free Arctic in our model selection compares well with similar criteria applied to CMIP6 models which predict the event between 2047 and 2052 while the process-based criteria advances the timing of the first ice-free summer up to 2035 (Docquier and Koenigk, 2021). However, the investigation of model selection criteria is out of scope of this study; our goal is to give an insight into when the Arctic might turn ice-free.



Our results highlight the increasing role of the MIZ in the response of Arctic sea ice to climate change. We show that
the MIZ will be the dominant sea ice class in the Arctic by 2050 which implies the shift to new sea ice conditions similar to
those in Antarctica. The chaotic interannual variability of the summer MIZF in the last years of simulations points out that the
current models' physics might not be suitable to changing sea ice conditions (Figure 10). In order to realistically simulate
(thermo)dynamical processes, the new sea ice regime requires modifications in the models' physics and sea ice rheology which
is formulated for thick pack ice (Aksenov et al., 2017). Additionally, the growing fraction of the MIZ requires changes in the
parameterization of the lateral and basal melt (Smith et al., 2022). The proper simulation of MIZ is essential for achieving
reasonable projections of future sea ice conditions since small and thin ice floes within the MIZ are more vulnerable to external
dynamic and thermodynamic forces than consolidated pack ice. In addition, the water patches between the ice floes permit the
absorption of solar radiation in the upper ocean, increasing the role of the ice-albedo effect which causes anticipation of the
ice-advance onset and acceleration of the overall sea ice loss. To demonstrate positive feedback between summer MIZ and
minimum SIA for the following year we plot the mean MIZF over June, July, August, and September (JJAS) against September
SIA with a 1-year lag computed for the years 2015-2050 (Figure 11a). All models except one simulate negative regression
ranging from $\sim$ -0.13 %/$10^6$ km$^2$ to -0.06 %/$10^6$ km$^2$ which means that the larger summer MIZF leads to lower September SIA
the following year. We suggest that the MIZ might act as a predictor of future sea ice conditions in the model simulations.
Figure 12b shows JJAS MIZF in 2015 (start of highres-future run) against the first September when the Arctic becomes ice-
free. Note that not all models simulate the event before 2050. Our analysis indicates that with the higher initial MIZF, the
September sea ice disappears earlier. This points out that the reasonable representation of the MIZ at the beginning of the run
might impact the pace of sea ice loss and potentially improve the accuracy of model projections. We assume that the MIZF
might represent a robust criterion to examine the model fidelity. The impact of the MIZ on the accuracy of the model
simulations needs further investigation.


## 5 Conclusions

In this study, we evaluate the historical and future variability of the Arctic sea ice area and volume using six coupled
atmosphere-ocean general models participating in the HighResMIP experiments of the sixth phase of the Coupled Model
Intercomparison Project (CMIP6). For the period 1979-2014, we find that most models can properly simulate maximum and
minimum of the SIA seasonal cycle at hemispheric and regional scales. However, some of them cannot correctly capture their
magnitude, failing to realistically reproduce the ice growth and retreat phases with systematic over- or underestimation of the
seasonal variability. We find that the models are generally able to reproduce the seasonal cycle of the Arctic-wide MIZ area,
although not all of them can capture the timing of the annual maximum. The models simulate different areas of the MIZ,
especially in summer, however, there is stronger agreement among models for MIZF. We find different regional contributions
to the inter-model spread associated to seasonal variability: the winter inter-model spread in SIA is attributed to the Atlantic



sector (Barents-Kara Seas and the Greenland ice zones), while the summer differences are tied to the the Laptev, East Siberian,
and Beaufort-Chukchi Seas.
Selected models broadly differ on the spatial distribution of the mean SIT as well as its average values. Only few models reveal
a pattern similar to PIOMAS characterised by thicker ice off the coast of Greenland and the Canadian Archipelago. Most
models simulate too thick ice which affects the representation of sea ice volume: excluding one outlier, all but two models
overestimate ice volume all year round up to 1.5 times in April and 3.5 times in August. However, regardless of large systematic
biases, most models simulate a realistic seasonal cycle of SIV with a maximum in April and a minimum in August. All models
capture declines in SIA and SIV over the historical period but they disagree on the pace of sea ice loss. The response to the
external forcing does change with season and region: the winter trends are dominated by changes in the Barents-Kara Seas and
the Greenland ice zone, while the summer trends are driven by those in the East Siberian, and Beaufort-Chukchi Seas. Most
models underestimate ice loss in all regions particularly in summer; conversely, they tend to simulate more negative trends  in
the Greenland zone leading to overestimating the Arctic-wide SIA trend in some configurations. In this study, we find that
there is no strong relationship between ocean/atmosphere resolution and sea ice cover representation: the impact of horizontal
resolution rather depends on the studied variable and the model used. However, the ocean has a stronger effect than the
atmosphere and the increase in the ocean resolution from $\sim 1°$ to $\sim 0.25°$ has a favourable impact on the representation of SIA
and sea ice edges which is especially evident for ECMWF-IFS and HadGEM3 models. At the same time, the simulation of
SIT does not directly rely on the grid spacing, as well as the derived SIV. A finer ocean resolution leads to lower SIV for
ECMWF-IFS and to almost no differences for HadGEM3. Increasing resolution both in ocean and atmosphere results in little
difference between configurations in CNRM and higher SIV for EC-Earth3P. On the other hand, enhanced atmosphere
resolution leads to higher SIV for ECMWF-IFS and CMCC-CM2 and lower SIV for HadGEM3 and MPI-ESM. We also find
that the difference between configurations varies from one region to another which highlights the importance to examine the
model performance at the regional scale. For example, CMCC-CM2 HR4 has too low SIA and SIV in the Barents Sea caused
by overestimating the OHT at the Barents Sea Opening (Docquier et al., 2020) while performing well in the rest of the sectors.
On the other hand, MPI-ESM has similar SIA in two configurations in the Barents-Kara Seas and the Greenland ice zone,
whereas the finer atmosphere configuration displays less sea ice in summer in the rest of regions.
Considering the period 2015-2050, all models simulate a long-term decrease in SIA and SIV with a generally stronger rate of
ice loss compared to the historical period. Model simulations predict that the Arctic loses nearly 95% of SIV from 1950 to
2050. T There is again no systematic impact of horizontal resolution on the occurrence of first ice-free conditions. The multi-
model mean of all models does not project the Arctic to become ice-free before 2050. However, applying the model selection
based on historical performance advances the event up to 2047. Considering that the model selection leads to closer agreement
with CDR on the year of first ice-free summer in the regions where it already happened (the East Siberian, Barents and Kara,
and the Laptev Sea), we infer that model selection application may potentially improve the accuracy of model projections of
Arctic sea ice evolution. Together with the overall ice shrinking, we studied the changes in the structure of sea ice cover and
we concluded that the MIZ will constitute up to 60-80% of the September SIA by 2050. This suggests a shift to a new sea ice





regime similar to that in the Antarctic. Given that the MIZ will play a major role in the response of the Arctic sea ice to external
forcing, modifications in the model physics and parametrizations are encouraged in the new generations of coupled climate
models.

**Author contributions**
JS and DI contributed to the conception and design of this study, JS made the analysis and wrote the manuscript. FC revised
the manuscript.
**Competing interests**
The contact author has declared that none of the authors has any competing interests.
**Acknowledgments**
JS and DI were supported by the European Union's Horizon 2020 research and innovation programme under grant agreement
No 101003826 via the project CRiceS. FC was supported by the Foundation Euro-Mediterranean Center on Climate Change
(CMCC, Italy).

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

**Figures**

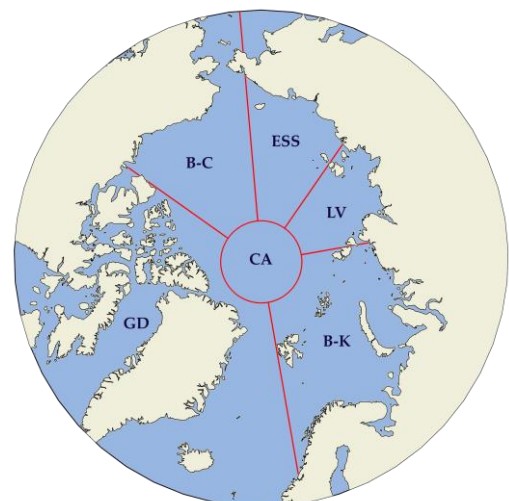

**Figure 1: Map of sub-regions used in the regional analysis: Central Arctic Basin (CA), Barents and Kara Seas (B-K), Laptev Sea (LV), East Siberian Sea (ESS), Beaufort and Chukchi Seas (B-C), Canadian Arctic Archipelago and Greenland coast (GD).**



a)

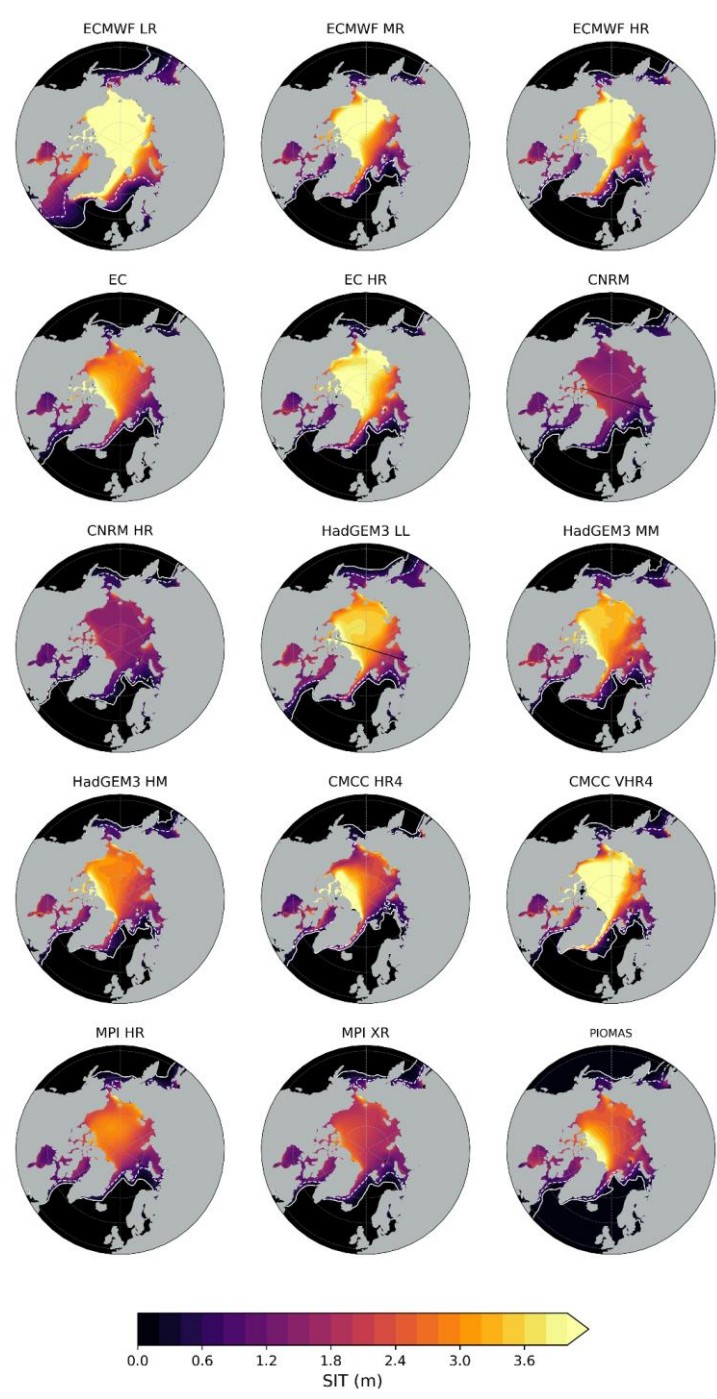



b)

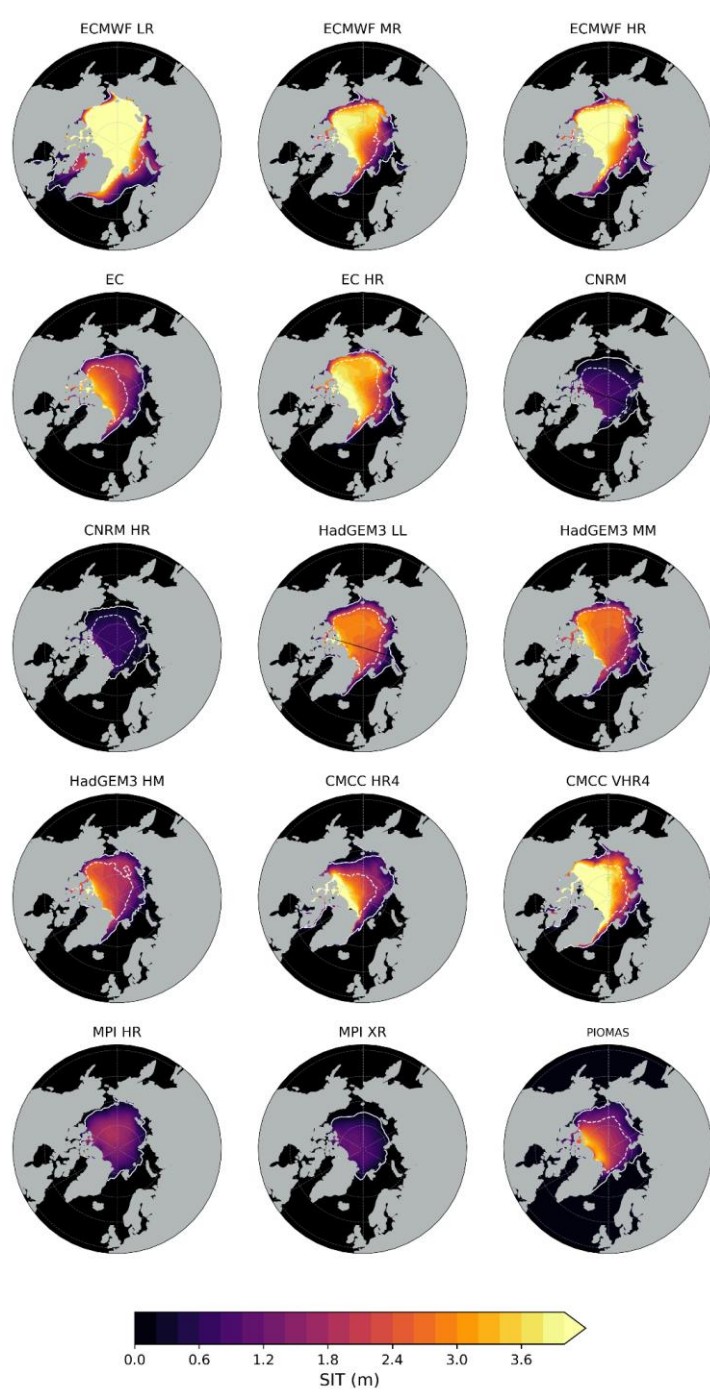

**Figure 2: The 1979-2014 climatological mean sea ice thickness from the model outputs and PIOMAS in March (a) and September (b). White contours show the edges of 15% (solid) and 80% (dashed) sea ice concentration from each model. SIC from CDR is used for PIOMAS.**




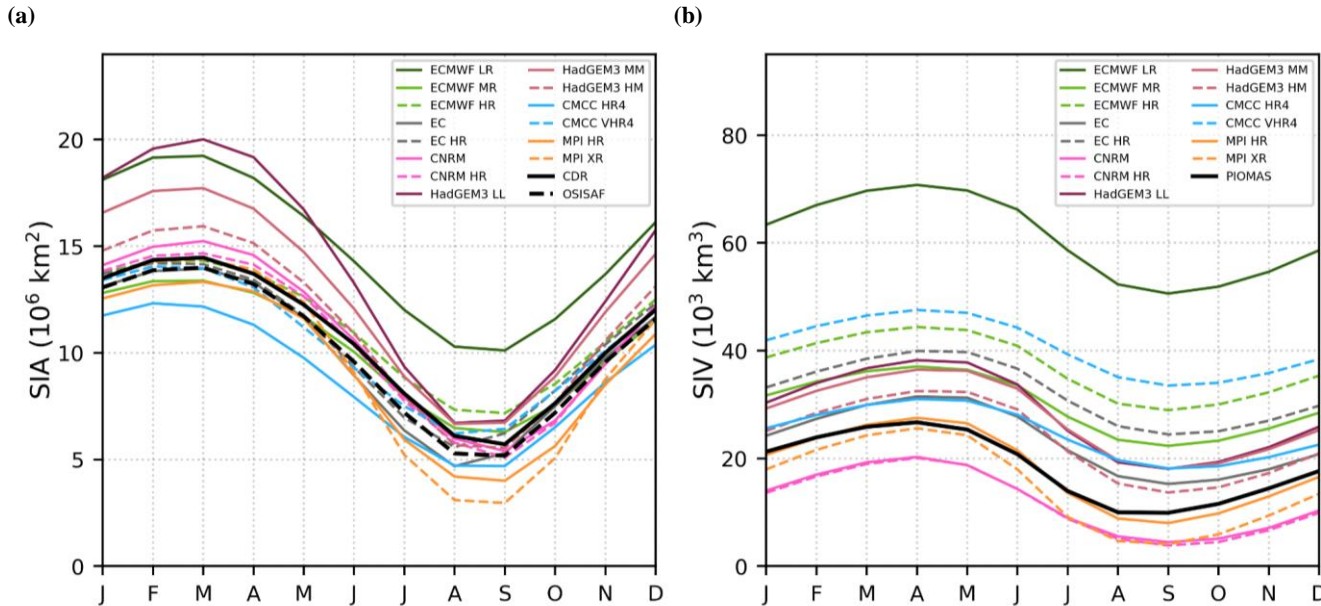

**Figure 3: The 1979-2014 seasonal cycle in SIA (a) and SIV (b) from HighResMIP hist-1950 model outputs against CDR and OSISAF for SIA and PIOMAS for SIV.**

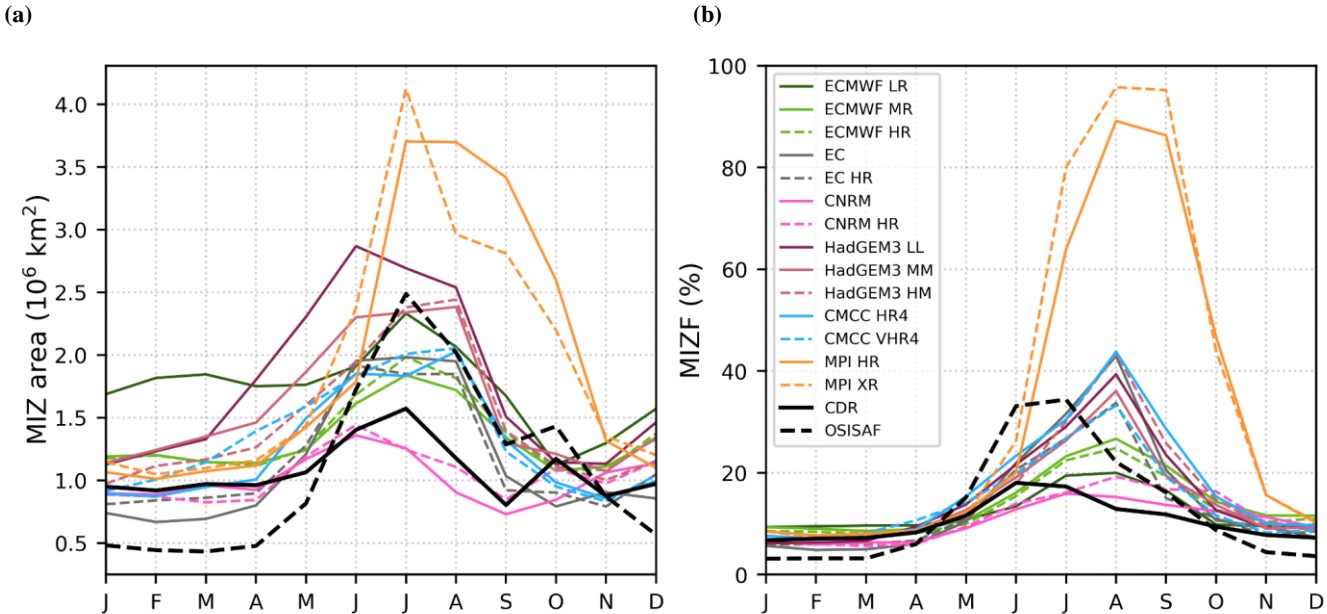

**Figure 4: The 1979-2014 seasonal cycle in the MIZ area (a) and MIZF (b) from HighResMIP hist-1950 model outputs and satellite products.**





**(a)**

b)

**Figure 5: The 1979-2014 seasonal cycle in a) SIA and b) SIV in the Arctic sub-regions from HighResMIP hist-1950 model outputs against CDR and OSISAF for SIA and PIOMAS for SIV.**





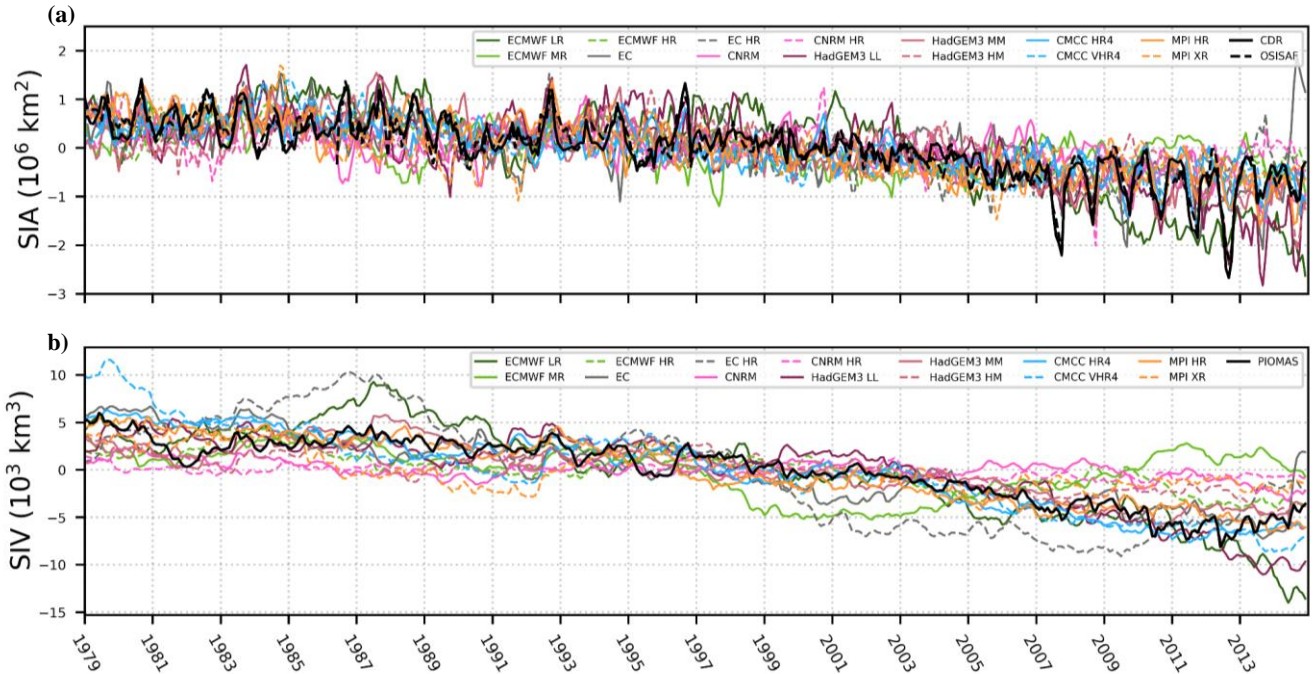

**Figure 6: Monthly anomalies of SIA (a) and SIV (b) over 1979-2014 from HighResMIP model outputs and reference products.**



**(a)**

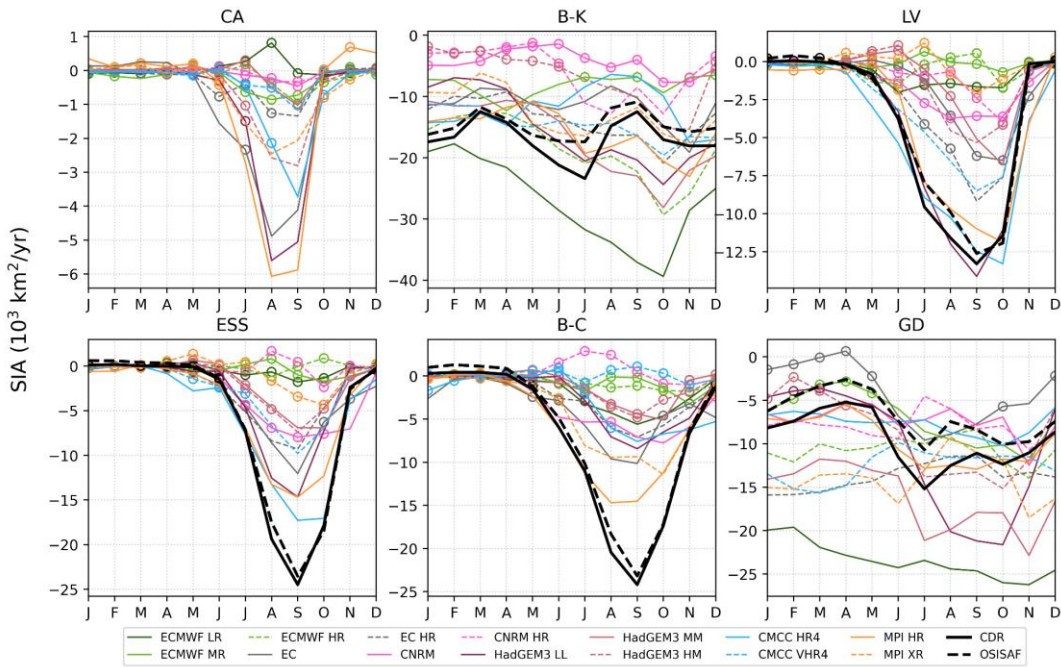

**b)**

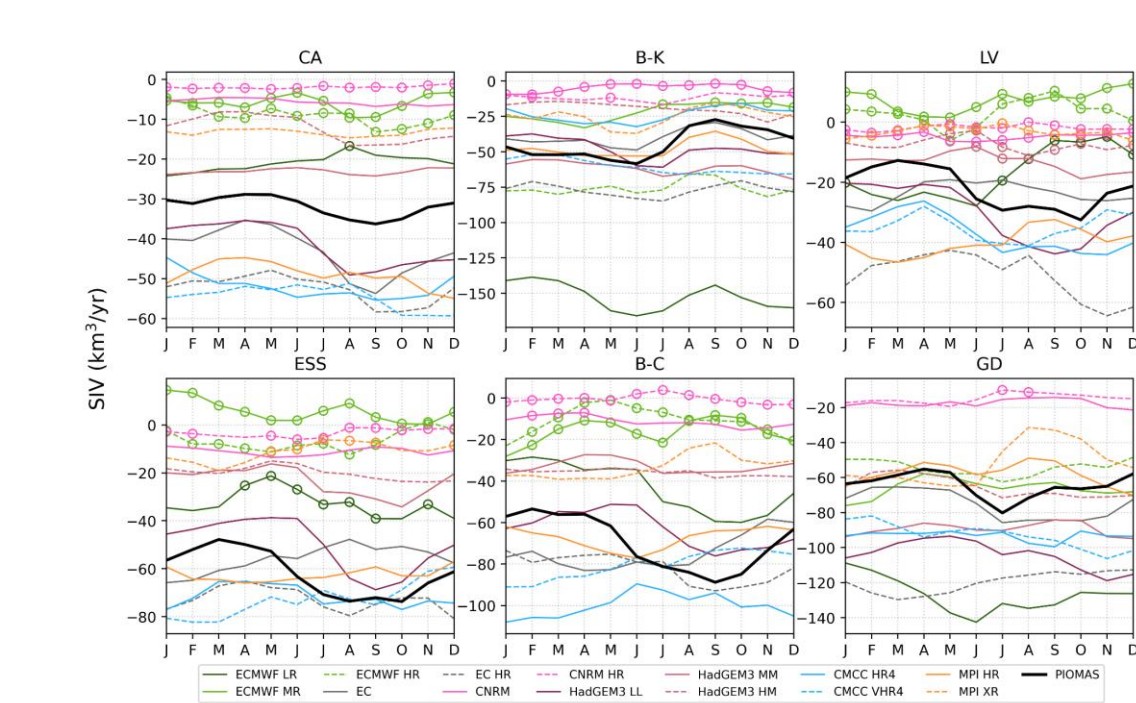

**Figure 7: The 1979-2014 monthly trends in SIA (a) and SIV (b) in the Arctic sub-regions for HighResMIP hist-1950 model outputs against CDR and OSISAF for SIA and PIOMAS for SIV. Dots indicate non-significant trends.**




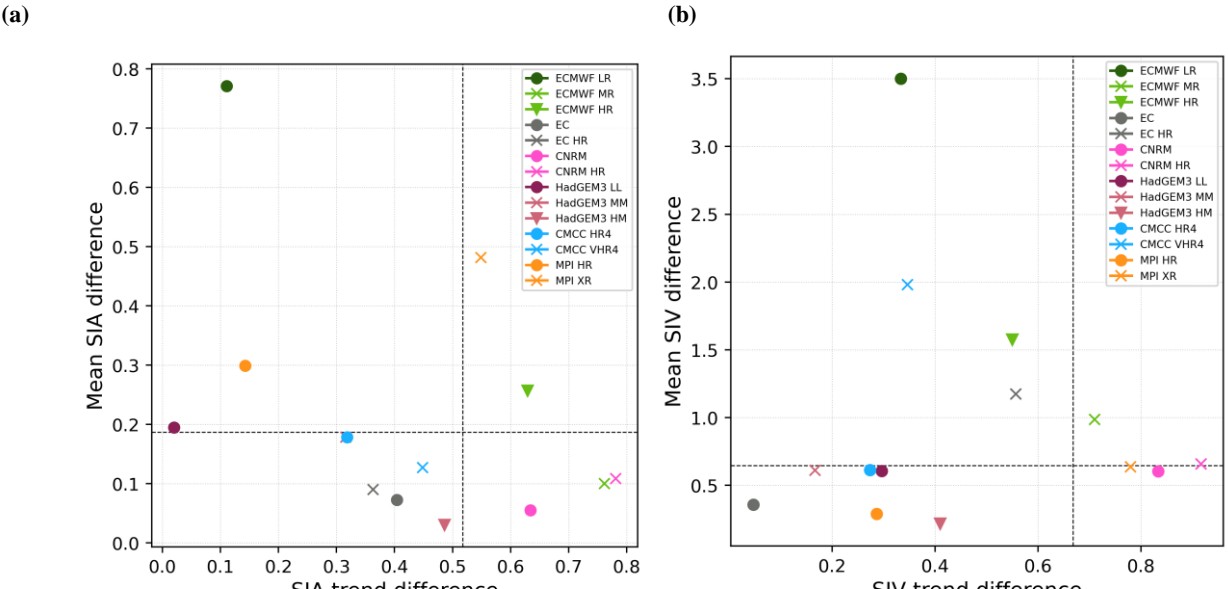

**Figure 8.** Normalized difference in mean September SIA against September SIA trend over 1979-2014 (a). Same for SIV (b). The difference is computed with reference to CDR (for SIA) and PIOMAS (for SIV). Dashed lines indicate 75th percentile for a set of the model outputs excluding ECMWF-IFS.



**Figure 9: Time series of September SIV from 1950 to 2050 using HighResMIP historical and future runs and PIOMAS for the entire Arctic and sub-regions. The multi-model mean SIV with model selection is shown by dashed line. The vertical lines indicate the time of ice-free conditions: green colour for the multi-model mean without model selection, yellow for the multi-model mean with model selection, and black for CDR. Free-ice conditions signify that SIA falls below $10^6$ km$^2$ for the total Arctic and reaches 25% of the CDR SIA averaged over 1980-2010 for the sub-regions.**



**(a)**

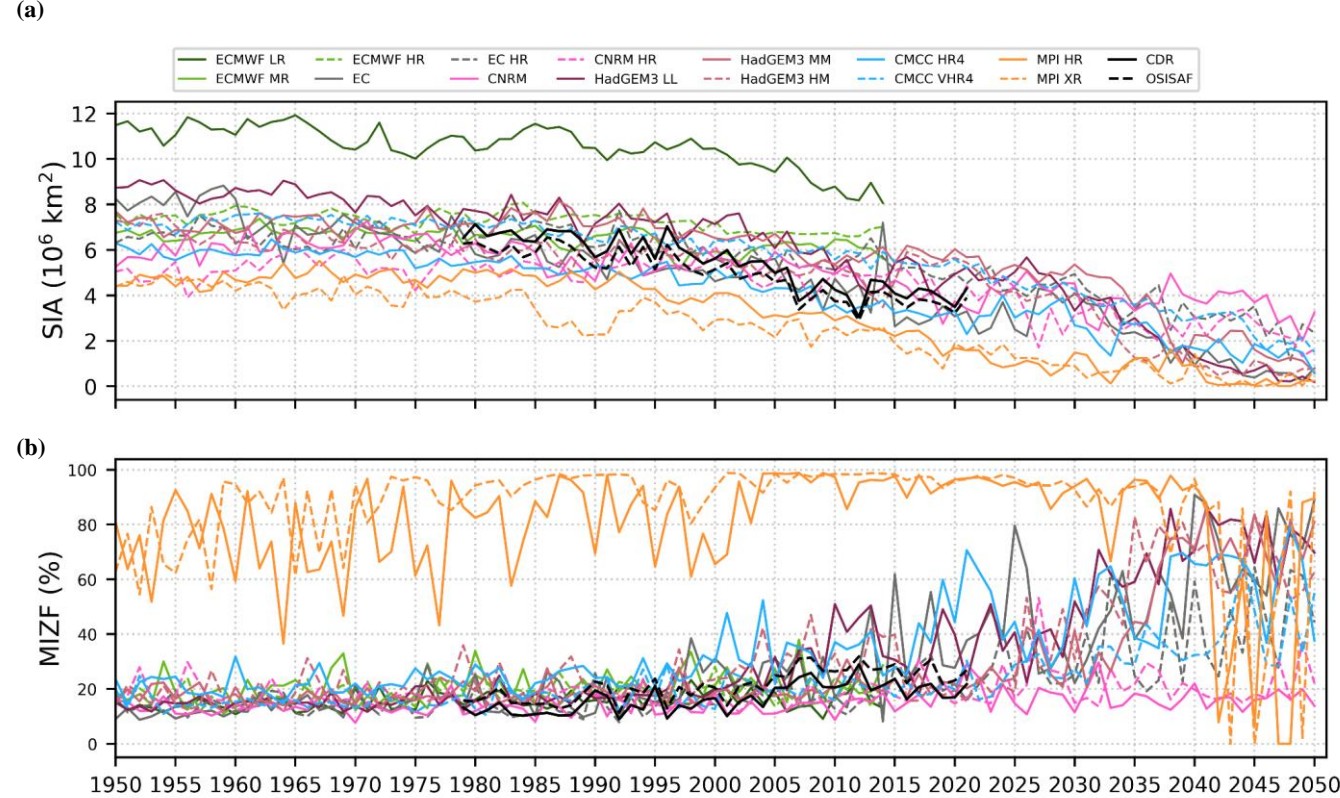

**Figure 10: Time series of September SIA (a) and MIZF (b) from 1950 to 2050 using HighResMIP historical and future runs and satellite products (CDR and OSISAF).**

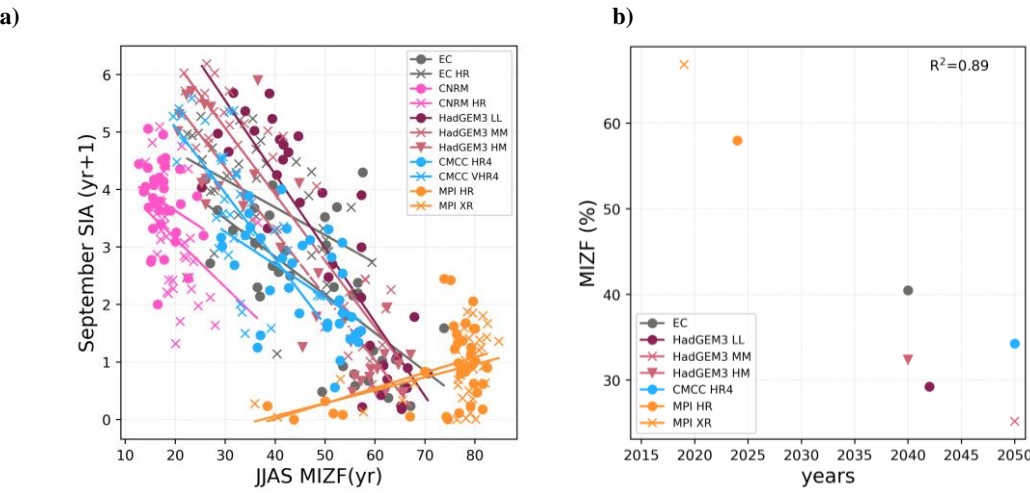

**Figure 11: June, July, August, and September (JJAS) MIZF mean against September SIA with one year lag over 2015-2050 (a); Timing of first ice-free Arctic against JJAS MIZF in 2015 (b).**