# Peer review of "Past and future of the Arctic sea ice in HighResMIP climate models"

_EGUsphere, 2023_

## Author Response (AR1)

Dear Editor and Reviewer,

We would like to thank you for your constructive comments and helpful suggestions, which substantially improve the quality of the manuscript. Answers to your comments are given in detail hereafter.

Reviewer comments are in blue, and are followed by our response (in black) that includes changes and/or additions to the text. All authors agree with the modifications made to the manuscript.

**Review report, Reviewer 1**

Review Selivanova, Iovino, Cocetta: Past and future of the Arctic sea ice in HighResMIP climate models

General:

The authors use a sub-set of HighResMIP model (those that have been used as part of the EU-project PRIMAVERA) to investigate the effect on increasing ocean and atmosphere resolution on the representation of ice area and ice thickness/ volume in the Arctic and within Arctic sub-regions and their variability and past and future trends. The main finding is that increasing the resolution does not lead to a robust signal in the analysed sea ice parameters across the investigated models.

The article is generally clearly structured and easily understandable, and also the figures are easy understandable. Not all the presented results are entirely new, mainly because high resolution often does not show substantially changes in the results and thus studies based on historical and future simulations with "standard-resolution" CMIP6 models have shown comparable results. However, there are still sufficient new and interesting results, which would justify a publication after some additional analysis and clarifications.

**Major points:**

1. The authors are not discussing the role of internal variability, and how it might affect differences between low and high-resolution versions of the models. Since only one member of each model configuration is analysed, internal variability itself can lead to large differences in both mean values and trends (compare for example results from Swart et al. 2015, Jahn et al. 2016 using single-model large ensembles or Karami et al. 2023 looking at long term internal variability), and internal variability can probably explain quite a part of the differences between the low and high resolution versions - at least for some parameters and regions. Internal variability might also be a contributing factor to the opposite response to increasing resolution that has been found in the different models, where the authors so far state that the response to increasing resolution depends on the model.

To solve this problem, authors could either include more HighResMIP models into their

analysis to investigate if more models show the same response to increasing resolution, or they could include all existing model members of the models they used in their analysis to find a more robust response in each single model. At the least, the authors should use proper significant tests to find out if the differences that have been found between LR and HR versions of the models are statistically significant.

We agree with the reviewer that internal variability might explain part of the differences between low- and high-resolution configurations. To access the impact of internal variability, we provide an additional analysis based on the results of the ECMWF models. We use ECMWF runs because the LR and HR configurations have at least six ensemble members. Only the other two models (EC-Earth3P and CNRM) have multiple members, but the ensemble is limited to three members at most, which might be not sufficient to properly sample internal variability. Figure S1 presents the seasonal variability of sea ice area (SIA) and sea ice volume (SIV), Figure S2 - the interannual variability of March and September SIA and SIV, while the table shows linear SIA and SIV trends over the period from 1979 to 2014. This analysis shows evidence that the choice of the first individual member is not a large limitation of our study. For this ensemble size, variability between single members does not surpass the effect of horizontal resolution on the Arctic sea ice representation. While the spread between HR members is very small, it increases for LR ensemble spread in the interannual variability and linear trends, but the findings of the paper remain valid regardless of the member choice. ECMWF LR largely overestimates the mean seasonal cycle and produces stronger negative SIA and SIV trends compared to HR counterparts.

[Figure]

**Figure S1. The 1979-2014 seasonal cycle in SIA (a) and SIV (b) from ECMWF LR and HR ensemble means against CDR and OSISAF for SIA and PIOMAS for SIV**. **Thin lines represent 6 ensemble members for LR and HR configurations.**

[Figure]

**Figure S2.** Time series of September and March SIA (a, b) and SIV (c, d) from ensemble means of ECMWF LR and HR runs. Thin lines represent 6 ensemble members for LR and HR configurations.

**Table S1. Linear trend in SIA and SIV and their standard deviations for ensemble members of ECMWF LR and HR simulations for 1979-2014.**

|  | 1979-2014 SIA trend ($10^3$ km$^2$/yr) | 1979-2014 SIV trend (km$^3$/yr) |
|---|---|---|
| ECMWF-IFR LR1 | -72.08 ± 16.9 | -423.86 ± 68.3 |
| ECMWF-IFR LR2 | -44.68 ± 13.5 | -238.32 ± 82.8 |
| ECMWF-IFR LR3 | -134.19 ± 14 | -801.13 ± 65.3 |
| ECMWF-IFR LR4 | -92.77 ± 12 | -666.2 ± 34.3 |
| ECMWF-IFR LR5 | -40.13 ± 8.8 | -608.42± 64.6 |
| ECMWF-IFR LR6 | -94.85 ± 11.5 | -225.26 ± 63 |
| **ECMWF-IFR LR Ens** | **-80.3 ± 9.1** | **-493.85±49.2** |
| ECMWF-IFR HR1 | -36.66 ± 7.6 | -157.49 ± 34.4 |
| ECMWF-IFR HR2 | -40.78 ± 8.9 | -209.46 ± 50.9 |

| | | |
|---|---|---|
| ECMWF-IFR HR3 | -36.92 ± 8.1 | -260.7 ± 58.7 |
| ECMWF-IFR HR4 | -32.45 ± 6.2 | -157.08 ± 47.4 |
| ECMWF-IFR HR5 | -31.95 ± 13.7 | -88.3 ± 61.8 |
| ECMWF-IFR HR6 | -11.29 ± 8.7 | -32.03 ± 49.1 |
| **ECMWF-IFR HR Ens** | **-31.81 ± 5** | **-150.85 ± 22.8** |

We argue that a clear assessment of internal variability is not feasable in the context of this paper. ECMWF does not provide model outputs for the future runs and EC-Earth3P and CNRM have only three members which might be not sufficient to capture internal variability, yet. Unfortunately, we cannot overcome the limitation due to model data availability shared with most of the recent studies using CMIP6 HighResMIP. For example, Docquier et al. (2019), Docquier et al. (2022), Koenigk et al. (2019) use ensemble-means only for ECMWF model and the first member for other models. Huang et al. (2021), Belucci et al. (2021), Tsartsali et al. (2022) use only the first member for each model.

2. Linked to the internal variability: At certain places the manuscript gives the impression (but I hope this was only somewhat misleading formulated) that the authors expect that global coupled models should be able to represent observed extremes. For example, they state that none of the models is able to reproduce the observed sea ice minima in 2007 and 2012 (section 3.4). Due to internal variability, we can never expect that a coupled model will represent a certain extreme in a certain year in long-term historical and future simulations as performed in CMIP6 / HighResMIP. Also, in section 3.4, the authors correlate annual sea ice values from the models with the satellite data to investigate how well the interannual variability is represented. If this is really what has been done (if I misunderstood please clarify what really has been correlated), this analysis must be removed because for the same reason as historical coupled model simulations cannot reproduce observed extremes at the observed time, they cannot represent observed interannual anomalies. That the correlations for SIV nevertheless provide relatively high values might be due to similar trends in satellite data and models – in case no detrending has been done before the correlation (even this should be clarified).

Thank you for the comment, we agree with your point. In the section 3.4, we correlated monthly anomalies from the model outputs and the satellite data. No detrending was done before the correlation. The part on the correlation analysis is removed from the section 3.4.

3. I am a bit puzzled by the regional results in section 3.3. Figure 5 shows large differences (up to 10-15%) in sea ice area between models in e.g. CA, LV, ESS even in winter when we

would expect that ice concentration is rather close to 100%. To me it seems that the area of the same sub-region is of different size in the different models. If you look at Figure 5 you will see that the sea ice area in models often is not changing at all between late autumn and spring, which indicates that the maximum ice area for this region has been reached. And if not the max sea ice concentration in some of the models have been hard-coded to 90 or less percent, then the area of the region just has to be smaller in those models which show a much smaller sea ice area. This would be a major issue since then the ice areas/ volumes from the different models and between models and satellite data and maybe even between the same model in different resolution cannot be compared. For proper comparison all model and observational data sets should be interpolated to the same grid and it has to be ensured that land-sea masks are comparable between the models.

Please explain more in detail how you calculated the area of the regions and carefully check all results in section 3.3, and if necessary update them.

The sea-ice area is the product of sea-ice concentration and grid-cell area, summed over all grid cells north of XXN. The grid cell area is computed on the model native grid, without appling any interpolation scheme.We followed the same approach by Docquier et al. (2019, 2020) in a similar analysis of SIA from HighResMip simulations. All the models, except MPI-ESM, employ the tripolar ORCA grid. Both model configurations of MPI-ESM used in this study employ the same ocean/ice tripolar grid. Model grids at different resolutions from the same model show small differences. Only the HadGEM3 model has a slightly larger error since sea ice concentration is provided on the atmosphere grid. The difference in ocean area north of 40°N is 3.9% between HadGEM3-MM and HadGEM3-LL when the atmosphere grid is used (as also showed in Docquier et al., 2019). It might contribute to the high bias in the winter SIA representation in some regions.. About the CA region, the bias is observed only for MPI-ESM model and it can be attributed to the different (not ORCA-like) tripolar model grid.

Minor points:

1. The HighResMIP-protocol foresees that differences between high and low resolution versions should be kept to a minimum, however, some models include automatized adaptions of parameter values to the resolution, other model groups had to make some small modifications to make the model running properly at high resolution. Further, the LR-versions normally were tuned and the HR-versions not at all. While this is in line with the HighResMPI-protocol, an untuned model might lead to substantial net surface and top of the atmosphere net energy imbalances, which leads to drifts and could affect the response of the climate.

I suggest to shortly mention that these factors might also contribute to the response to increasing the resolution.

Thank you for your suggestion. We included it in the discussion. "And finally, part of the differences between the low and high resolutions might be explained by the parameter adaptation to the grid size and small modifications between different model configurations. Furthermore, the low-resolution configurations are more carefully tuned in CMIP exercise compared to high-resolution versions. That can lead to drifts and consequently impact the climate response".

2. Line 31-33: The sentence is formulated such as it sounds as the strongest trend is in September and the 2nd strongest in March. But I guess what is meant is that the strongest trend is in September and the lowest trend is in March. Please check and clarify.

Yes, the sentence was poorly formulated. We rephrased it (Line 33).

3. Line 50: '…no significant changes in the area of the Arctic MIZ… '' But the position of the MIZ changed? If yes, maybe worth to mention.

It is true - the position of the MIZ is changed (Rolph et al., 2018) but in this sentence, we stress out the difference between the absolute MIZ area and the proportion of the MIZ in the total SIA which implies changes in the average MIZ position.

4. Line 98/99: 'For the past, sea ice properties …' This sentence is badly placed here. I suggest to remove it here and mention the period else where (e.g. around line 112).

Thank you for this suggestion. The sentence is moved to the beginning of the next paragraph (Line 199-120).

5. Table 1: Since the focus is on sea ice, it might make sense to compare some basics of the sea ice model and sea ice parameterizations as well.

The setup and physical choises of the sea ice model and all other components of the climate system affect our results and conclusions. This comparison is anyway out of the scope of this study. Furthermore, detailed description of the sea ice model physics and parameterizations is not available for all model configurations and models.

6. Line 152: '…does not change significantly …'. Please clarify how you calculated the significance.

We corrected the sentence (Line 161).

7. Line 159: 'Configurations with finer ocean resolution have a better fit to CDR …' How do you estimated this? Bye eye? Except for the ECMWF-model, I have difficulties to clearly see this.

We agreed that the comparison was not clear enough. We removed the sentence.

8. Line 172: It seems that a number of models simulate despite too thick ice, too small ice concentrations compared with satellite data. I think this could be an important finding for

guiding sea ice model improvements and their tuning, and should be mentioned somewhere, e.g. in the discussion or conclusion section.

In the same context, see also Fig. 3, some models, particularly, both CMCC-versions, EC-Earth3P-HR and ECMWF simulate the ice edge pretty well (and the ice area) but SIT (and ice volume) is much too large. Normally, I would expect that too thick ice should be to a certain degree linked to too large ice areas as well but this does not really seem to be the case. Do you have any explanation for this piling up of sea ice in the Arctic? Sea ice model short comings? Too strong lateral melting? Or too strong Beaufort Gyre keeping all the ice in the Central Arctic?

Thank you for the comment. We do agree that the overtestimation of ice thickness can be attributed to a combination of thermodynamics and dynamics processes, and interplay with ocean/atmoshere system. We have here only analysed "classical" ice variables as distributed from the modeling centers. Analysis on the physical sources of differences, beside the changes in grid resolution, is not a main question of this study. We leave this investigation for further studies.

9. Line 184: Models tend to have the SIA minimum too early, several already in August. Maybe worth to mention.

Most models and the reference products have similar values in August and September but the annual minimum occurs in September.

10. Line 219-221: The two sentences are saying more or less the same. I suggest merging them to one, e.g.: It is worth noting that the evaluation of the simulated MIZ area is highly dependent on the reference product used, particularly in summer.

The sentences are merged into one single sentence following your suggestion (Line 230-231).

11. Line 292-295: It is not surprising that models agree in winter sea ice area better than in summer ice area in Arctic Ocean areas since in winter the Arctic Ocean is entirely ice covered. In summer, winter/ spring thickness is one important factor for how quickly the area is getting ice free and certainly differences between models in summer atmospheric circulation and temperature play a major role as well. I am sure heat input (and volume input) of rivers also matters in reality, however, I am not aware, that any of the models explicitly simulates temperature of the inflowing river water. Do not most models use the ocean temperature at the river mouth/ ocean points where the runoff enters the ocean? If you would like to state that the heat input from rivers is the main reason for the summer spread across models, please provide more details on how the different models differ in terms of their river representation and how large the difference in terms of heat-input into the Arctic from rivers is between models.

Thank you for your comment. The river schemes among these climate models are different and can employ different parameters and physical complexity. In deep analysis of river input

is not included into this study also because we do not have detailed informations and output from this component of any single runs. Our speculation is not scientifically based on numerical results, then we rephrased the sentence. "This could be possibly associated with the model differences in simulating atmospheric circulation, as well as representation of the river discharge (Park et al., 2020) and the transport of Pacific waters through the Bering Strait (Watts et al., 2021), which modify the thermo-haline structure of the upper-ocean and affect sea ice growth and melt" (Line 306-307).

12. Section 3.3: The rather lengthy description in this section is partly a bit difficult to follow. A table - for at least SIA - showing winter (March) and summer (September) ice area for each region in each model and in satellite data would help.

We add the table in the section 3.3 as you suggested (Line 354).

13. Line 349: EC-Earth3P-HR simulates much thicker ice than the LR-version. This is likely explaining why the trend in ice area is smaller.

Thank you for you comment. We added it into the text "Here, the exception is EC-Earth3P in which the eddy-permitting configuration has a larger negative trend in SIV (-322 and -460 km3 yr−1). This might be attributed to the thicker ice simulated in HR configuration (Figure 2)" (Line 385).

Technical corrections:

Line 98: Add '8.5' to SSP5-scenario: SSP5-8.5.

Line 130 'north' instead of 'North'

Line 170: 'addition' instead of 'Addition'

Line 408: I suggest to change '14 models' with '14 simulations' , since only 6 different models are used.

All technical corrections are done.

Dear Editor and Reviewer,

We would like to thank you for your constructive comments and helpful suggestions, which substantially improve the quality of the manuscript. Answers to your comments are given in detail hereafter. We hope that you will find them satisfactory.

Our responses to each of the reviewer's comments are provided below in black.

All authors agree with the modifications made to the manuscript. Reviewer comments are in blue, and are followed by our response (in black) that includes changes and/or additions to the text.

Julia Selivanova, Doroteaciro Iovino, Francesco Cocetta

Selivanova et al., examine past and future Arctic sea-ice variability and changes in a subset of models participating in HighResMIP. The authors investigate how high-resolution (HR) and low-resolution (LR) versions of a model affect a range of Arctic sea ice variables, including sea ice thickness, volume, area, and concentration both over the historical record and under a future emissions scenario. The authors find that increasing the horizontal model resolution does not lead to a any significant difference in Arctic sea ice.

This manuscript is clear and the figures are quite clear. Despite the main result showing that HR models does not substantially change Arctic sea ice trends (when compared to LR models), it is important to document and has important implications for future modeling efforts with refined grids. However, I think there are some overlooked aspects of this result that might change the key message. Thus, I think this manuscript should be published after some additional analysis and clarifications. Below I describe these concerns and suggestions.

**Major**

I am concerned that the authors overlooked the role of internal variability on Arctic sea ice trends and variability. Internal variability is known to be highly model dependent (Bonan et al., 2021) and strongly influence sea ice trends (Swart et al., 2015). I think it would be helpful for the authors conduct additional analyses that examine other members of each model. A quick glance at the HighResMIP archive (https://esgf-node.llnl.gov/search/cmip6/) suggests this is possible for at least some models. For instance, CNRM-CM6-1 has 10 members. If not all models have more ensemble members, it could be worthwhile to focus on comparing HR and LR results in a model with 10 ensemble members (e.g., CNRM-CM6.1). My belief is that HR and LR models will have different "forced" responses and this results itself could broaden the study. I also think the HR and LR models will likely have different internal variabilities based on Fig. 3 which shows that HR and LR models have different SIV mean states.

In summary, I strongly suggest the authors conduct additional analyses that essentially repeat this analysis but with a more robust quantification of the "forced" response and internal variability.

We thank the referee for this relevant comment.We agree with the reviewer that internal variability might explain part of the differences between low- and high-resolution configurations. To access the impact of internal variability, we provide an additional analysis based on the results of the ECMWF models. We use ECMWF runs because the LR and HR configurations have at least six ensemble members. Only the other two models (EC-Earth3P and CNRM) have multiple members, but the ensemble is limited to three members at most, which might be not sufficient to properly sample internal variability. Figure S1 presents the seasonal variability of sea ice area (SIA) and sea ice volume (SIV), Figure S2 - the interannual variability of March and September SIA and SIV, while the table shows linear SIA and SIV trends over the period from 1979 to 2014. This analysis shows evidence that the choice of the first individual member is not a large limitation of our study. For this ensemble size, variability between single members does not surpass the effect of horizontal resolution on the Arctic sea ice representation. While the spread between HR members is very small, it increases for LR ensemble spread in the interannual variability and linear trends, but the findings of the paper remain valid regardless of the member choice. ECMWF LR largely overestimates the mean seasonal cycle and produces stronger negative SIA and SIV trends compared to HR counterparts.

[Figure]

**Figure S1. The 1979-2014 seasonal cycle in SIA (a) and SIV (b) from ECMWF LR and HR ensemble means against CDR and OSISAF for SIA and PIOMAS for SIV**. **Thin lines represent 6 ensemble members for LR and HR configurations.**

[Figure]

**Figure S2.** Time series of September and March SIA (a, b) and SIV (c, d) from ensemble means of ECMWF LR and HR runs. Thin lines represent 6 ensemble members for LR and HR configurations.

**Table S1.** Linear trend in SIA and SIV and their standard deviations for ensemble members of ECMWF LR and HR simulations for 1979-2014.

| | 1979-2014 SIA trend ($10^3$ km$^2$/yr) | 1979-2014 SIV trend (km$^3$/yr) |
|---|---|---|
| ECMWF-IFR LR1 | -72.08 ± 16.9 | -423.86 ± 68.3 |
| ECMWF-IFR LR2 | -44.68 ± 13.5 | -238.32 ± 82.8 |
| ECMWF-IFR LR3 | -134.19 ± 14 | -801.13 ± 65.3 |
| ECMWF-IFR LR4 | -92.77 ± 12 | -666.2 ± 34.3 |
| ECMWF-IFR LR5 | -40.13 ± 8.8 | -608.42± 64.6 |
| ECMWF-IFR LR6 | -94.85 ± 11.5 | -225.26 ± 63 |
| **ECMWF-IFR LR Ens** | **-80.3 ± 9.1** | **-493.85±49.2** |
| ECMWF-IFR HR1 | -36.66 ± 7.6 | -157.49 ± 34.4 |
| ECMWF-IFR HR2 | -40.78 ± 8.9 | -209.46 ± 50.9 |
| ECMWF-IFR HR3 | -36.92 ± 8.1 | -260.7 ± 58.7 |

| ECMWF-IFR HR4 | -32.45 ± 6.2 | -157.08 ± 47.4 |
|---|---|---|
| ECMWF-IFR HR5 | -31.95 ± 13.7 | -88.3 ± 61.8 |
| ECMWF-IFR HR6 | -11.29  ± 8.7 | -32.03 ± 49.1 |
| **ECMWF-IFR HR Ens** | **-31.81 ± 5** | **-150.85 ± 22.8** |

We argue that a clear assessment of internal variability is not feasable in the context of this paper. ECMWF does not provide model outputs for the future runs and EC-Earth3P and CNRM have only three members which might be not sufficient to capture internal variability, yet. Unfortunately, we cannot overcome the limitation due to model data availability shared with most of the recent studies using CMIP6 HighResMIP. For example, Docquier et al. (2019), Docquier et al. (2022), Koenigk et al. (2019) use ensemble-means only for ECMWF model and the first member for other models. Huang et al. (2021), Belucci et al. (2021), Tsartsali et al. (2022) use only the first member for each model.

**Minor**

L33 and L38: Cite Fetterer et al., 2016 and Stroeve & Notz, 2018 instead of the https links.

We provide trends over 1979-2022 and 1979-2021 that cannot be cited in the papers dated the years 2016 and 2018.

Line 170: Remove capital "A" from addition.

L184-185: It is probably worth mentioning that some models have biases as their summer minimum is in August rather than September like in observations.

Most models and the reference products have similar values in August and September but the annual minimum occurs in September.

L407-409: I would suggest changing models to simulations. Only six models were used.

All minor corrections are done.

---

## Author Response (AR2)

Reviewer 1:
The authors have made the requested revisions based on the previous round of review. I have
no further substantive comments here. I therefore recommend that this paper be accepted.

We appreciate that the reviewer has found our revised manuscript acceptable for publication.

Reviewer 2:
The authors have answered all points to my satisfaction except for one.
This point is about the internal variability.
While I understand that it is difficult to completely assess the role of internal variability due
to missing ensemble members, I still think it is important to use the information that is
available on internal variability to potentially strengthen the conclusions on potential impact
of the high-resolution models versus internal variability. As long as the difference between a
high-resolution model version and its lower-resolution comparison is in the range of internal
variability of two members from the same model version, the conclusions are not robust.
To use 3 ensemble members is thus already substantially better than only using 1 member.
Second, even for one member (or for the three), one could apply any kind of appropriate
significant test to strengthen the conclusions – or to come to the conclusion that more
ensemble members would be needed in the future. Also, this could be an important finding
for the high-resolution model community.
The supplementary figure on the 6 ECMWF-model members provides interesting results. It
shows that there is a very large variability in the sea ice trends. And while I agree with the
authors that for the ECMWF model, the resolution has a robust impact on sea ice and its
trends, we have to have in mind that the ECMWF model shows by far the largest effect of
high resolution. Differences between high and low resolution for the other models are much
smaller and for most models smaller than the spread across the ECMWF-LR or HR members.
This means that we cannot be really sure that the difference is due to the change in resolution.
Thus, I suggest to add at least a similar table as S1 for the 3 members of the other two models
with 3 members.
At the very least, a proper discussion of internal variability and its potential impacts on the
results and the uncertainty connected to it is should be added to the Discussion or Conclusion
section.

We thank the reviewer for raising this important issue. The role of internal variability in
climate change has been documented in many studies, with a significant impact on Arctic ice
decline (e.g. Ding et al., 2019) and also the timing of an ice-free Arctic (e.g. Wettstein and
Deser, 2014). Large ensembles of multidecadal simulations are needed to adequately sample
internal climate variability and identify model deficiencies and strengths. We acknowledge
that these are computationally expensive and may only be sometimes available.
The role of internal variability in Arctic projections might be comparable to the effect of grid
resolution enhancement in sea ice representation.

Only few HighResMIP models provide a small number of members, others only one simulation. ECMWF-IFR LR, MR, and HR have 8, 3, and 6 ensemble members, respectively; CNRM and EC-Earth3P have 3 ensemble members for both the LR and HR systems. To address the reviewer's concerns, we provided additional analysis in the supplementary material and a new paragraph in section 4.

The potential effect of small ensembles on the differences between low and high-resolution model versions is shown by the SIA and SIV variability on seasonal (Figures S1, S3, and S5) and interannual (Figures S2, S4 and S6) timescales, and linear SIA and SIV trends (Table S1, S2 and S3) from 1979 to 2014 for ensemble members of LR and HR configurations. From seasonal variability, the differences between the ensemble members are very small and the effect of spatial resolution does not depend on the choice of the ensemble member. The impact of internal variability is larger on trends, but the variability between single members does not surpass the effect of horizontal resolution on the Arctic sea ice representation, and the ensemble means are generally comparable with the first-member trends shown in the manuscript. Multidecadal internal variability is very difficult to quantify from single simulation or small ensemble size. Accounting for fluctuations due to internal variability requires much larger ensembles that are yet not available within the HighResMIP framework. We think that our analysis provides some insight into the limited benefit of atmospheric and oceanic spatial resolutions, and explores the model fidelity in representing present and future sea ice state in the Arctic, while it also underlines the need for large ensembles of multidecadal simulations to strengthen our efforts towards developing more realistic climate models.

Ding, Q., Schweiger, A., L'Heureux, M. *et al.* Fingerprints of internal drivers of Arctic sea ice loss in observations and model simulations. *Nature Geosci* 12, 28–33 (2019). https://doi.org/10.1038/s41561-018-0256-8

Wettstein, J. J., and C. Deser, 2014: Internal Variability in Projections of Twenty-First-Century Arctic Sea Ice Loss: Role of the Large-Scale Atmospheric Circulation. J. Climate, 27, 527–550, https://doi.org/10.1175/JCLI-D-12-00839.1.